# Complex strain evolution of polar and magnetic order in multiferroic BiFeO$_3$ thin films

Zuhuang Chen[1,2], Zhanghui Chen[3], Chang-Yang Kuo[4,5], Yunlong Tang[2], Liv R. Dedon[2], Qian Li[6], Lei Zhang [2], Christoph Klewe[3,7], Yen-Lin Huang[2], Bhagwati Prasad[2], Alan Farhan [7], Mengmeng Yang[6], James D. Clarkson[2], Sujit Das[2], Sasikanth Manipatruni[8], A. Tanaka [9], Padraic Shafer [7], Elke Arenholz[7], Andreas Scholl[7], Ying-Hao Chu[10], Z.Q. Qiu[6], Zhiwei Hu[4], Liu-Hao Tjeng[4], Ramamoorthy Ramesh[2,3,6], Lin-Wang Wang[3] & Lane W. Martin [2,3]

Electric-field control of magnetism requires deterministic control of the magnetic order and understanding of the magnetoelectric coupling in multiferroics like BiFeO$_3$ and EuTiO$_3$. Despite this critical need, there are few studies on the strain evolution of magnetic order in BiFeO$_3$ films. Here, in (110)-oriented BiFeO$_3$ films, we reveal that while the polarization structure remains relatively unaffected, strain can continuously tune the orientation of the antiferromagnetic-spin axis across a wide angular space, resulting in an unexpected deviation of the classical perpendicular relationship between the antiferromagnetic axis and the polarization. Calculations suggest that this evolution arises from a competition between the Dzyaloshinskii–Moriya interaction and single-ion anisotropy wherein the former dominates at small strains and the two are comparable at large strains. Finally, strong coupling between the BiFeO$_3$ and the ferromagnet Co$_{0.9}$Fe$_{0.1}$ exists such that the magnetic anisotropy of the ferromagnet can be effectively controlled by engineering the orientation of the antiferromagnetic-spin axis.

[1] School of Materials Science and Engineering, Harbin Institute of Technology, Shenzhen 518055, China. [2] Department of Materials Science and Engineering, University of California, Berkeley, CA 94720, USA. [3] Materials Sciences Division, Lawrence Berkeley National Laboratory, Berkeley, CA 94720, USA. [4] Max-Planck Institute for Chemical Physics of Solids, Nöthnitzer Straβe 40, Dresden 01187, Germany. [5] National Synchrotron Radiation Research Center, 101 Hsin-Ann Road, Hsinchu 30076, Taiwan. [6] Department of Physics, University of California, Berkeley, CA 94720, USA. [7] Advanced Light Source, Lawrence Berkeley National Laboratory, Berkeley, CA 94720, USA. [8] Components Research, Intel Corp., Hillsboro, OR 97124, USA. [9] Department of Quantum Matter, ADSM, Hiroshima University, Higashi-Hiroshima 739-8530, Japan. [10] Department of Materials Science and Engineering, National Chiao Tung University, Hsinchu 300, Taiwan. These authors contributed equally: Zuhuang Chen, Zhanghui Chen. Correspondence and requests for materials should be addressed to Z.H.C. (email: zuhuang@hit.edu.cn) or to L.W.M. (email: lwmartin@berkeley.edu)

Antiferromagnetic materials play a critical role in the growing field of antiferromagnetic spintronics and more broadly, electric-field control of magnetism[1]. Regardless of the applications, knowledge of the antiferromagnetic-spin structure and manipulation of the spin axis are essential for both fundamental understanding of exchange interactions between antiferromagnetic and ferromagnetic layers and for enabling deterministic performance in spintronics[2,3]. But despite this fact, compared with ferromagnets, there are relatively few studies on controlling the spin structure of antiferromagnets mainly due to the limited probes of spin structure in these materials that lack macroscopic net magnetization.

Among antiferromagnets, BiFeO$_3$ is particularly interesting because it exhibits robust room-temperature multiferroism (ferroelectricity and antiferromagnetism)[4,5] and magnetoelectric coupling that allows one to use electric fields to manipulate magnetic order[6]. In turn, researchers have demonstrated the potential for electric-field control of ferromagnetism in exchange-coupled ferromagnet/BiFeO$_3$ heterostructures, making BiFeO$_3$ a prime candidate for low-power spintronics and nanoelectronics[7]. Despite this potential, BiFeO$_3$ is a complex material with many variables that must be controlled. For example, bulk BiFeO$_3$ has a rhombohedral $R3c$ structure with spontaneous polarization (**P**) and antiphase octahedral rotations along the $\langle 111 \rangle$ (pseudocubic indices are used throughout unless otherwise specified), which gives rise to complex domain patterns and switching pathways[8,9]. The magnetic order of bulk BiFeO$_3$ crystals is also complex, as it exhibits G-type antiferromagnetism with a superimposed long-wavelength cycloidal modulation along the $\langle 1\bar{1}0 \rangle$[10,11]; that is, the spins rotate within the $\{11\bar{2}\}$ containing the direction of the spontaneous polarization **P** and the cycloid modulation vector[12,13]. Antiphase oxygen octahedra rotations permit canting of the antiferromagnetic lattice through the Dzyaloshinskii–Moriya interaction (DMI) resulting in a local, weak canted moment **M**[14,15], while the spin-cycloid structure results in the cancellation of net macroscopic magnetization and linear magnetoelectric coupling in bulk BiFeO$_3$[16–19]. This said, it is reported that epitaxial constraints in thin films can suppress the spin cycloid[20–22], and drive a transition toward a homogenous, weakly-ferromagnetic order with a preferred antiferromagnetic spin axis (**L**) in $\{111\}$, which is perpendicular to the oxygen octahedral rotation axis and the direction of **P** (Fig. 1a)[6,7,23,24]. This can be understood using a phenomenological Hamiltonian consisting of a DMI term and a spin–spin exchange interaction term[23]. The DMI term has the form $E_{DM} = -\mathbf{D}\cdot(\mathbf{L}\times\mathbf{M})$, where $E_{DM}$ is the DMI energy and **D** is the DM vector. Due to symmetry arguments, **D** is determined by the sense of rotation of the oxygen octahedra and is thus oriented along $\langle 111 \rangle$[23]; i.e., parallel to **P**. A perfect antiferromagnetic order as preferred by the exchange interaction term will have zero DMI energy, while canting of the magnetic moment can make the DMI energy negative, and the most efficient way to have such canting and to reach the lowest energy is when **L** is perpendicular to **D** (and, in turn, **P**)[23]. Previous studies, mainly focused on (001)-oriented BiFeO$_3$ films, reported that compressive strain drives **L** to exhibit the largest out-of-plane component while remaining in the plane perpendicular to **P** (i.e., $[11\bar{2}]$); while tensile strain favors **L** lying in the film plane (i.e., $[1\bar{1}0]$)[24]. Other studies found a more complex strain evolution of the antiferromagnetic-spin structure, including reports that **L** tends to orient along in-plane $[1\bar{1}0]$ at large compressive strains and along the out-of-plane $[001]$ under tensile strains[25]. Regardless of the approach, such analyses are complicated by the existence of multiple ferroelastic domains in (001)-oriented films, which could be partially responsible for the diversity of observations[24,25]. Finally, despite these observations, the mechanism responsible for the strain-induced spin structure

change has not been well developed; precluding further understanding and control of magnetism in BiFeO$_3$-based heterostructures.

Here, we employ angle- and polarization-dependent soft X-ray absorption spectroscopy (XAS) and X-ray linear dichroism (XLD) together with computational approaches to investigate the influence of epitaxial strain on the antiferromagnetic-spin structure in (110)-oriented BiFeO$_3$ thin films. We demonstrate from both experiment and theory that, while epitaxial strain has relatively little impact on the orientation of **P**, it can drive a continuous reorientation of **L** from in-plane to out-of-plane directions over a wide angular space, such that **P** and **L** are no longer perpendicular when the films are under tensile strain. Our calculations suggest that spin–spin exchange coupling and the DMI dominate at low strain, giving rise to a perpendicular relationship between **P** and **L**. At large strain values, however, the single-ion anisotropy (SIA) increases in magnitude until it is comparable with the DMI, leading to a gradual deviation of the perpendicular relationship between **P** and **L**. Leveraging the strong exchange coupling between the ferromagnet Co$_{0.9}$Fe$_{0.1}$ and BiFeO$_3$, we demonstrate that the anisotropy of the ferromagnet can be tuned by 90° by controlling the strain state of the BiFeO$_3$. Ultimately, this work provides new insights on the origin of magnetic anisotropy in BiFeO$_3$ films, demonstrates a pathway to break the expected perpendicular relationship between **P** and **L**, and provides fundamental understanding to enable controllable tuning of spin orientation in BiFeO$_3$-based heterostructures.

## Results

**Growth and characterization of (110) BiFeO$_3$ films with a single structural domain.** To understand the effect of epitaxial strain on the antiferromagnetic-spin structure of BiFeO$_3$, 12–70 nm thick BiFeO$_3$ films were grown via pulsed-laser deposition on SrTiO$_3$ (110) and GdScO$_3$ (010)$_O$ (where the subscript O denotes orthorhombic indices) substrates such that (110)-oriented films are produced (Methods and Supplementary Fig. 1). (110)-oriented films were chosen to reduce the domain variants in BiFeO$_3$ such that only one **P** and **L** projection on the (110) is possible[8]. For brevity, we focus on four heterostructure variants encompassing two representative BiFeO$_3$ thicknesses: 12-nm-thick films (which are coherently strained to both the SrTiO$_3$ and GdScO$_3$ substrates) and 70-nm-thick films. These show anisotropic strain relaxation such that the films are coherently strained only along the $[001]$ ($[001]_O$) and relaxed along the $[1\bar{1}0]$ ($[100]_O$) for growth on SrTiO$_3$ (GdScO$_3$) (Supplementary Fig. 2). Off-axis reciprocal space mapping (Fig. 1b and Supplementary Fig. 2) and piezoresponse force microscopy (Supplementary Fig. 3) studies show that the films are monodomain.

**Study of antiferromagnetic spin axis via XLD.** The magnetic structure was probed with XLD which arises from two different origins: magnetic-linear dichroism and crystal-field-induced linear dichroism[26]. In BiFeO$_3$, temperature-dependent XLD studies have found that the intensity of the XLD signal near $T_N$ is much smaller than at 300 K, especially for XLD at the Fe-$L_2$ edge which essentially vanishes (Supplementary Figs. 4 and 5); indicating that the XLD in BiFeO$_3$ is largely dominated by a magnetic origin[27]. Representative pairs of XAS (taken in normal incidence with the polarization vector **E** of the incoming X-rays parallel to the $[001]$ (blue curves) and $[1\bar{1}0]$ (red curves); Fig. 1c) and XLD spectra (Fig. 1d) taken on 70-nm-thick BiFeO$_3$ heterostructures reveal an opposite polarization dependence and a reversal of linear dichroism between films grown on GdScO$_3$ and SrTiO$_3$ substrates. The Fe $L_{2,3}$ XAS consists of two absorption peaks because of multiplet effects (denoted as $A$ and $B$; Fig. 1c), and the spectral

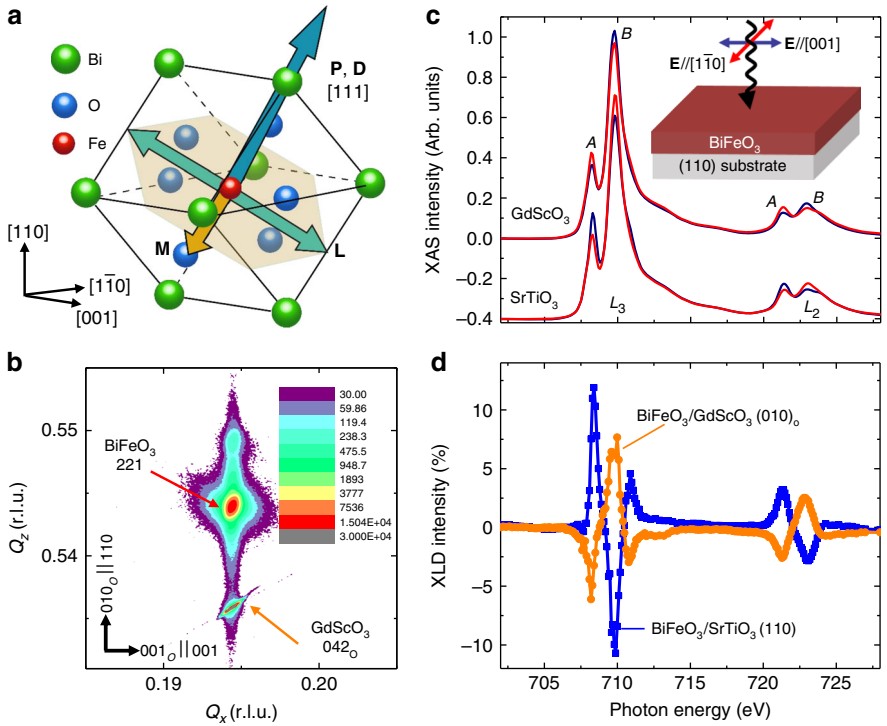

**Fig. 1** Strain-induced change in magnetic anisotropy in monodomain BiFeO₃ films. **a** Schematic of a (110)-oriented BiFeO₃ crystal structure wherein the polarization **P**, oxygen octahedral rotation axis **D** (blue arrow), easy-magnetic plane (orange hexagon), antiferromagnetic-spin axis **L** (red arrow), and canted moment **M** (orange arrow) are drawn. **b** Example reciprocal space mapping result, here for a 70-nm-thick BiFeO₃/GdScO₃ (010)$_O$ heterostructure, about the 042$_O$- (221-) diffraction conditions of GdScO₃ (pseudocubic BiFeO₃). No peak splitting is observed, indicating that the film has single ferroelastic domain structure. **c** Polarization-dependent X-ray absorption spectroscopy (XAS) and **d** X-ray linear dichroism (XLD) spectra at normal incidence about the Fe $L_{2,3}$ edge taken on 70-nm-thick BiFeO₃/GdScO₃ (010)$_O$ and SrTiO₃ (110) heterostructures

shape depends on the relative orientation of **E**, the crystal-lographic axes, and **L**[28]. The reversed dichroism (Fig. 1d) suggests that the orientation of **L** for these two heterostructures is different.

To better understand the reason for and nature of these changes in the XLD for the different heterostructures, we completed angle- and polarization-dependent XAS studies with various incident X-ray directions for the BiFeO₃ films on both SrTiO₃ and GdScO₃ substrates. The relative angle $\theta$ between **E** and a specified crystallographic axis is varied by rotating the samples about the X-ray Poynting vector with different incident angles of the X-rays (Fig. 2a). The $L_{2A}/L_{2B}$ peak intensity ratios, calculated from the XAS as a function of $\theta$, were used to extract the magnetic nature of the films. The $L_{2A}/L_{2B}$ ratio exhibits a strong angle and polarization dependence (Fig. 2b–e), suggesting a uniaxial anisotropy (i.e., an easy axis) in the BiFeO₃ films, since an easy-plane magnetic structure would give rise to a smaller dichroism and weaker angle and polarization dependence[24]. In addition, the $L_{2A}/L_{2B}$ ratios show a markedly different trend with the polarization angle for the four different film variants (Fig. 2b–e); indicating that strain effectively modifies the orientation of **L**.

To extract the orientation of **L** for the different heterostructure variants, we have simulated the experimental XAS spectra using configuration interaction cluster calculations (Supplementary Figs. 6 and 7)[29]. For the 12-nm BiFeO₃/GdScO₃ heterostructures (coherently strained to the substrate with a compressive strain of −2% along the [1̄10] and a tensile strain of 0.1% along [001̄]), the experimental results (points, Fig. 2b) can be well reproduced by simulations with **L** along the in-plane [1̄10] (lines, Fig. 2b). For the 70-nm BiFeO₃/GdScO₃ heterostructures (relaxed and strained along the [1̄10] and [001̄], respectively), the experimental results

(points, Fig. 2c) can be well reproduced by simulations with **L** along the out-of-plane [110] (lines, Fig. 2c). For the 12-nm BiFeO₃/SrTiO₃ heterostructures (coherently strained to the substrate with a compressive strain of −1.4% along both the [1̄10] and [001̄]), the experimental results (points, Fig. 2d) can be reproduced by simulations with **L** along the in-plane [001̄] (lines, Fig. 2d). Finally, for the 70-nm BiFeO₃/SrTiO₃ heterostructures (relaxed and strained along the [1̄10] and [001̄], respectively), the experimental results (points, Fig. 2e) can be reproduced by simulations with **L** rotated from the in-plane [001̄] by ~35° in the out-of-plane direction (solid lines, Fig. 2e). What these analyses suggest is that, in BiFeO₃, **L** is highly sensitive to the strain state of the material and gradually reorients from in-plane to out-of-plane directions over a wide angular space with increasing tensile strain.

**First-principles calculations**. To understand the fundamental origin of this strain-driven reorientation of **L**, we performed density functional theory (DFT) calculations (Methods) to explore the evolution of both **L** and **P** under different strains. For each strain state, we calculate the energy landscape when **L** points in different directions (Fig. 3a). In a strain-free film, we find that **L** is energetically degenerate within a plane perpendicular to **P** (here we set **P** to be along the [111]); consistent with previous studies[23]. Under compressive strain, we find that the easy-plane degeneracy is gradually lifted resulting in **L** continuously rotating to point along the in-plane [1̄10] at large compressive strains. Under tensile strain, the easy-plane degeneracy also disappears and **L** is gradually changed as the axis first converges to point approximately along the [112̄]. Upon further increasing the magnitude of the tensile strain, **L** rotates toward the out-of-plane [110]. The change of **L** with strain is summarized (red lines,

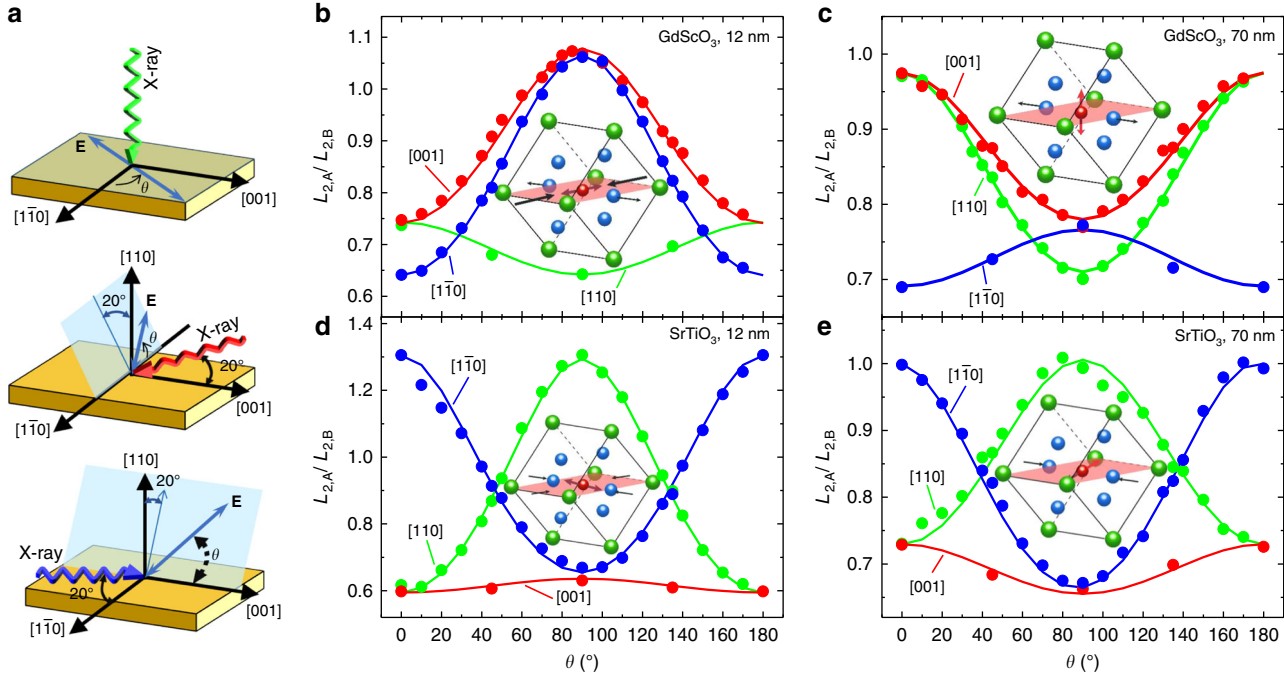

**Fig. 2** Angle- and polarization-dependent X-ray absorption spectroscopy. **a** Schematics of the measurement geometries where $\theta$ is the angle between the polarization vector **E** of the incoming X-rays (blue arrows) and the principal crystallographic axes as labeled. Experimental (solid points) and calculated (lines) show the polarization dependence of the Fe-$L_2$-peak ratio as a function of $\theta$ for incident X-rays along the [110] (green lines), [001] (red lines), and [1$\bar{1}$0] (blue lines) of **b** 12 nm and **c** 70 nm thick films on GdScO$_3$ (010)$_O$, and **d** 12 nm and **e** 70 nm thick films on SrTiO$_3$ (110). Insets of (**b–e**) show the schematics of the strain states and the orientation of **L** (red arrow)

Fig. 3b). Note that the DFT results are calculated at 0 K and the magnetic anisotropy energies (sub meV) are at the resolution of the DFT calculations[29,30]. Thus, as is common practice, a relatively large strain is used to demonstrate the trend, which agrees well with the experiments.

Having observed that **L** rotates in a continuous manner as the strain is varied, this begs the question: Is **P** also rotating considerably during application of strain to maintain the classical perpendicular coupling to **L**? To explore this, we have evaluated **P** using the Berry-phase method under different strain conditions (Fig. 3c) and find that the in-plane $\mathbf{P}_{[1\bar{1}0]}$ component is maintained essentially constant at ~0 C/m$^2$ while the perpendicular, in-plane $\mathbf{P}_{[001]}$ and out-of-plane $\mathbf{P}_{[110]}$ components increase and decrease in magnitude, respectively, as the strain varies from compressive to tensile in nature. This suggests that **P** rotates only by an amount of ±15° from the [111] within the (1$\bar{1}$0) (blue lines, Fig. 3b). Importantly, this gradual strain-driven rotation of **P** is not synchronized with the rapid rotation of **L** (Fig. 3a, b). As a result, **P** and **L** will no longer be perpendicular under some stain states and the angle between **P** and **L** will vary nonlinearly with strain starting from 90° and reaching a minimum of 46° at large tensile strains (Fig. 3d). To confirm that the strain-driven polarization rotation is not synchronized with the rotation of **L**, we completed polarization mapping with scanning transmission electron microscopy (STEM and Methods) (data shown here for the 70 nm BiFeO$_3$/GdScO$_3$ heterostructure, Fig. 3e). Magnified bright-field images (Fig. 3f) allow for direct measurement of the cation and anion displacements. Polarization maps were produced by extracting these displacements and reveal a uniform polarization direction close to the expected [111] (Fig. 3g and Supplementary Figs. 8 and 9). This observation is further supported by polarization-electric field hysteresis loop measurements, which reveal that the out-of-plane **P** of the (110)-oriented films is ~90 µC/cm$^2$ (Supplementary Fig. 10), as expected[31].

Ultimately, this confirms that **P** and **L** can deviate markedly from the classically expected perpendicular configuration.

Previous studies on the magnetic structure of BiFeO$_3$ have generally considered only the DMI contribution which is related to the Fe–O–Fe bond angle (inset, Fig. 4a), while the SIA, which is related to the distortion of the FeO$_6$ octahedra (i.e., Bi–Fe distance[30]), has been less studied. Using an ab initio derived spin Hamiltonian (Methods and Supplementary Fig. 11), we find that the SIA is very sensitive to the strained-induced lattice distortion and that the change of **L** with strain results from a change of the balance between the DMI (as represented by **D**, Fig. 4a) and the SIA (as represented by the SIA constant $K$, Fig. 4b) effects[32]. When the DMI energy dominates over the SIA energy, the **L** and **P** will be approximately perpendicular, otherwise, this relationship will be broken. Closer inspection of the trends reveals a number of important observations. At zero strain, $K$ is more than 10-times smaller than **D**[30]; therefore, the magnetic anisotropy is dictated by the DMI and thus **L** is predicted to be constrained within an easy plane perpendicular to **D** and, ultimately, **P**. It has been reported that when $K$ becomes larger than a critical value, a simple G-type antiferromagnet becomes robust against an incommensurate magnetic structure with an easy-magnetic plane[30,33]. This is consistent with our calculations wherein the disappearance of the easy-magnetic plane in the strained films can be explained by the enhancement of the $K$ value via strain-induced structural distortion. $K$ is dramatically enhanced in films under large compressive or tensile strain and becomes comparable in magnitude to **D**. At large compressive strains, $\mathbf{D}_{[1\bar{1}0]}$ is essentially zero, and $\mathbf{D}_{[110]}$ is considerably smaller than $\mathbf{D}_{[00\bar{1}]}$; suggesting that **D** is aligned along the [00$\bar{1}$] (Fig. 4a). Because **D**, **L**, and **M** form a right-handed coordinate relationship, **L** will prefer to remain in the (001). Furthermore, even though $\mathbf{D}_{[110]}$ is small, the fact that it is non-zero breaks any degeneracy and leads

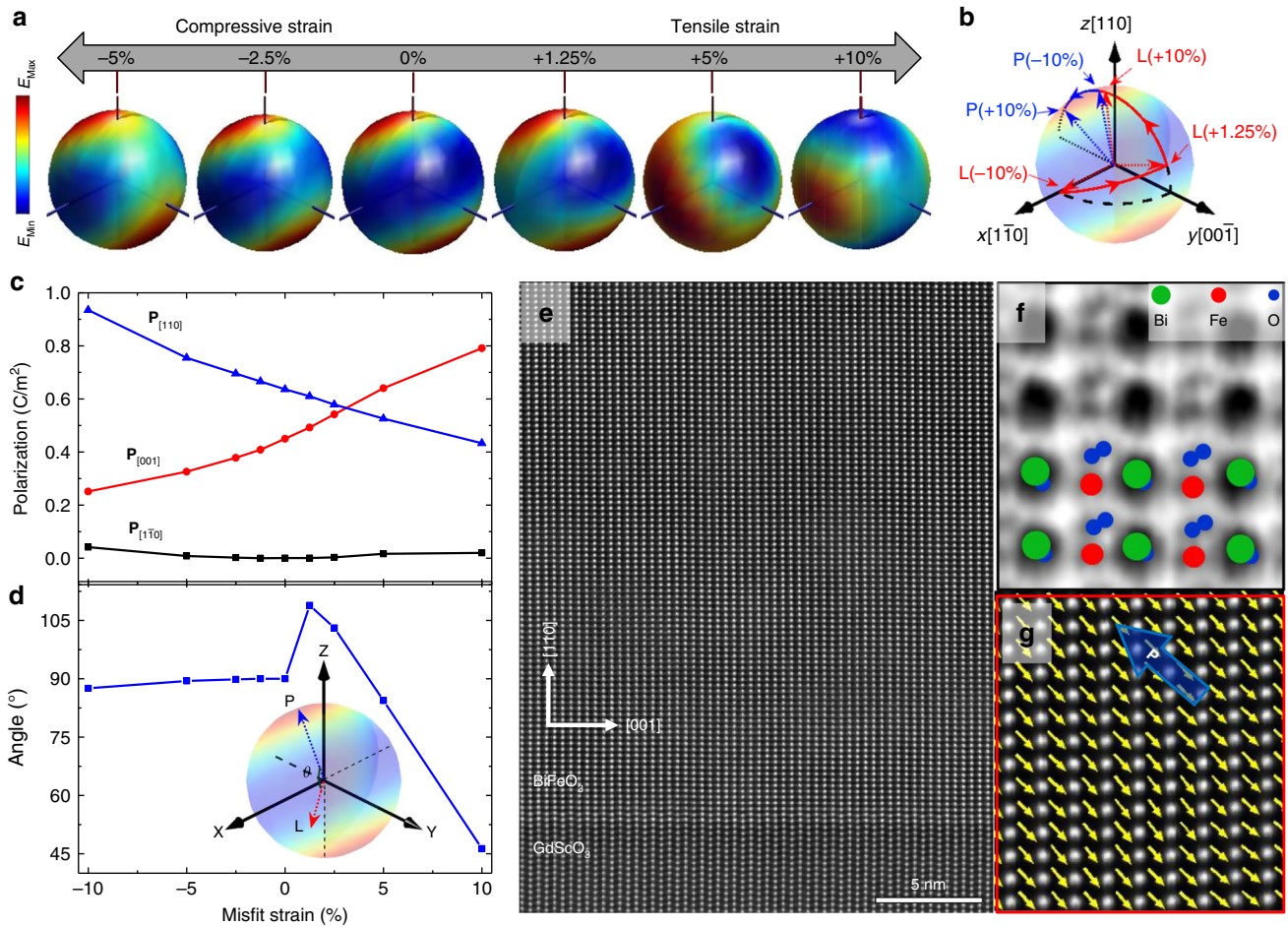

**Fig. 3** Evolution of antiferromagnetic-spin and polarization with strain. **a** Three-dimensional representation of the evolution of the magnetic energy landscape under various strain states. The color in the representation indicates the total energy when the antiferromagnetic-spin axis **L** points in that direction wherein cool and warm colors represent low and high energy, respectively, and thus easy and hard axes, respectively. **b** Summary of the evolution of antiferromagnetic-spin axis **L** (red arrow) and polarization **P** (blue arrow) with strain from first-principles calculations. **c** The three components of the polarization as a function of misfit strain from the first-principles calculations. **d** The angle $\theta$ between the polarization vector and the antiferromagnetic-easy axis as a function of misfit strain. **e** High-resolution high-angle annular dark field-scanning transmission electron microscopy (HAADF-STEM) image of a 70 nm $BiFeO_3$/$GdScO_3$ $(010)_O$ heterostructure. **f** Bright field-scanning transmission electron microscopy (BF-STEM) image of the same heterostructure showing the Fe cation and O anion displacement details. Direct imaging of all species in the BF-STEM image enables direct extraction of the polar distortions. **g** Fe-cation displacement vector maps relative to its two neighboring Bi cations; the films are found to exhibit uniform structural distortion as noted, indicating that the polar direction of the film is uniform and close to the expected $\langle 111 \rangle$

to **L** preferring to point along the in-plane $[1\bar{1}0]$. This is augmented by the SIA since, in the case of compressive strain, the out-of-plane Fe–O bonding strength is weakened while the in-plane term is enhanced, which results in the SIA further driving **L** towards the in-plane $[1\bar{1}0]$[34]. At large tensile strains, however, things are more complicated. $\mathbf{D}_{[1\bar{1}0]}$ is again essentially zero, and $\mathbf{D}_{[00\bar{1}]}$ is now marginally larger than $\mathbf{D}_{[110]}$, indicating that **D** should be within the $(1\bar{1}0)$, and is driven closer to the $[00\bar{1}]$, thus **L** tends to be in the $(00\bar{1})$. Note that $K$ (~40 μeV) is also comparable to **D** (<90 μeV). Thus under tensile strain, the out-of-plane Fe–O bonding strength is enhanced while the in-plane Fe–O bonding is weakened (Supplementary Fig. 12), meaning that $K$ will rotate out-of-plane, resulting in **L** pointing along the out-of-plane $[110]$ rather than in-plane $[1\bar{1}0]$[34]. **P** and **D** point to the same $[111]$ under zero strain, but they deviate under strain. This is because **D** is related to the Fe–O–Fe bonding angle while **P** is related to the relative displacement of the positive (Bi, and to a lesser extent, Fe) and negative (O) charge centers which experience only minor changes (Supplementary Fig. 13) under

strain; thus leading to the gradual rotation of **P** and its asynchrony with the rotation of **L**. The fine sensitivity of SIA to structural distortion may also explain how even small misfit strains in $BiFeO_3$ films can be sufficient to suppress the spin cycloid[20–22].

**Controlling magnetic anisotropy of a coupled ferromagnetic layer.** Having developed a picture of the fundamental nature of magnetic structure evolution with strain and the underlying mechanism for those changes, we continue to probe the effect of strain-induced antiferromagnetic spin reorientation on the exchange coupling with a ferromagnet. 2.5 nm Pt/2.5 nm $Co_{0.9}Fe_{0.1}$ heterostructures were grown on the 70-nm-thick $BiFeO_3$ films under a field ($H_g = 200$ Oe) applied either along the in-plane $[00\bar{1}]$ or $[1\bar{1}0]$ (Methods). Representative magneto-optical Kerr effect (MOKE) hysteresis loops taken from the $Co_{0.9}Fe_{0.1}$/$BiFeO_3$ heterostructures where $H_g$ was applied along the $[00\bar{1}]$ (Fig. 5a, b) and the $[1\bar{1}0]$ (Supplementary Fig. 15) illustrate that, irrespective of the orientation of $H_g$, the ferro-magnetic easy axis is always along $[00\bar{1}]$ and $[1\bar{1}0]$ for the

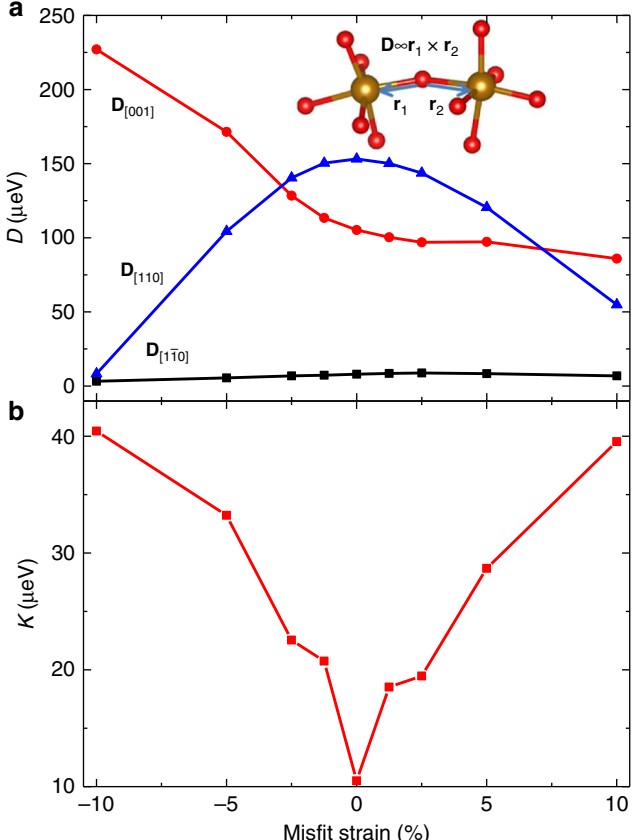

**Fig. 4** Magnetic evolution with strain via spin Hamiltonian. **a** Strain dependence of the in-plane ($[1\bar{1}0]$, $[00\bar{1}]$) and out-of-plane ($[110]$) components of the **D** vector. Inset shows a schematic of the Dzyaloshinskii–Moriya interaction between two Fe cations (gold) connected by oxygens (red). **b** Strain dependence of the single-ion-anisotropy constant $K$.

ferromagnetic interface is due to the spin-flop coupling mechanism. According to the spin-flop mechanism, it is energetically preferred for a small canted moment in the antiferromagnet to couple parallel to the ferromagnetic magnetization, giving rise to a uniaxial magnetic anisotropy in the ferromagnet[38]. The XLD measurements have also shown that the **L** of the 70 nm $BiFeO_3$/$GdScO_3$ heterostructures rotates towards the out-of-plane direction. Thus the in-plane anisotropy is reduced, consistent with magnetic torque results which reveal that the in-plane uniaxial anisotropy constant in heterostructures grown on $SrTiO_3$ is much larger than those on $GdScO_3$ (Fig. 5d). All told, our studies demonstrate that a strong exchange coupling between the ferromagnetic $Co_{0.9}Fe_{0.1}$ and antiferromagnetic $BiFeO_3$ exists such that the anisotropy direction of the $Co_{0.9}Fe_{0.1}$ is set by the orientation of **L** of the underlying $BiFeO_3$ layer.

**Conclusions**. In summary, we demonstrate the ability to tune the antiferromagnetic-axis orientation from in-plane to out-of-plane across as wide angular space on (110)-oriented $BiFeO_3$ thin films via epitaxial strain. A deviation of the classical perpendicular relationship between the antiferromagnetic axis and the polarization vector was found in both experiments and ab initio calculations. This phenomenon arises due to the interplay of the DMI and SIA effects. By engineering the antiferromagnetic-spin orientation, in turn, we can effectively tune the magnetic anisotropy of exchange-coupled $Co_{0.9}Fe_{0.1}$ layers. Our results enable a deeper understanding of the magnetic nature of $BiFeO_3$ and exchange interaction at the $BiFeO_3$/ferromagnet interface, and will help to design next-generation spintronic devices, such that electric-field control of magnetic spin orientation may be more readily achieved.

## Methods

**Sample preparation**. Epitaxial $BiFeO_3$ thin films were grown on $SrTiO_3$ (110) and $GdScO_3$ $(010)_O$ single-crystal substrates via pulsed-laser deposition at 680 °C in a dynamic oxygen pressure of 100 mTorr[39]. Following growth, the $BiFeO_3$ films were cooled in ~700 Torr of oxygen to room temperature at a rate of 5 °C/min. Detailed structural information was obtained using high-resolution X-ray diffraction (X'Pert MRD Pro, Panalytical) including $\theta$–$2\theta$ scans and reciprocal space maps (RSMs).

**Soft X-ray absorption spectroscopy**. X-ray spectroscopy measurements were carried out at beamline 4.0.2 of the Advanced Light Source (ALS) at Lawrence Berkeley National Laboratory and beamline 08B of the National Synchrotron Radiation Research Center (NSRRC) in Taiwan. The measurements were performed in total-electron-yield (TEY) geometry. The XLD measurements were obtained from the difference of horizontal and vertical polarized light absorption spectra. The sample temperature was controlled using an in-vacuum resistive heater. After heating to 600 K, we checked the XAS at 300 K again and found no surface degradation caused by the heating (Supplementary Fig. S5). The X-ray beam was incident on the sample at an angle of 20° and 90° from the sample surface for grazing incidence and normal incidence, respectively; the light polarization was selected using an elliptically-polarizing undulator. Spectra were captured with the order of polarization rotation reversed (e.g., horizontal, vertical, and then vertical, horizontal) so as to eliminate experimental artifacts. The angle- and polarization-dependent XAS measurements were independently performed at room temperature for the $BiFeO_3$ films using TEY geometry at NSRRC. The relative angle $\theta$ between light polarization vector **E** and the crystallographic axes are varied by rotating the samples about the X-ray Poynting vector. Such experimental geometry allows for the X-ray penetration path length of the incoming beam to be independent of the polarization angle $\theta$, guaranteeing a reliable comparison of the spectral line shapes as a function of $\theta$[40].

**Configuration interaction cluster calculations**. To extract the orientation of the antiferromagnetic axes of the various (110)-oriented $BiFeO_3$ films, we have simulated the experimental spectra using configuration interaction calculations with the octahedral $FeO_6$ cluster, based on atomic multiplet theory and the local effects of the solid[41,42]. It takes into account the intra-atomic $3d$–$3d$ and $2p$–$3d$ Coulomb and exchange interactions, the atomic $2p$ and $3d$ spin-orbit coupling of the Fe ion, the oxygen $2p$–Fe $3d$ hybridization, and the octahedral crystal-field of Fe $3d$ orbital interaction[43]. The simulations were carried out using the program XTLS 8.3, and the parameters used in the $FeO_6$ cluster calculations for $BiFeO_3$ film are:

heterostructures grown on $GdScO_3$ (Fig. 5a) and $SrTiO_3$ (Fig. 5b), respectively. Without the $BiFeO_3$ layer, the easy axis of the $Co_{0.9}Fe_{0.1}$ film is always along the $[00\bar{1}]$ on the two substrates; again irrespective of the direction of $H_g$ (blue and green curves, Fig. 5a, b, and Supplementary Fig. 16). All $Co_{0.9}Fe_{0.1}$/$BiFeO_3$ heterostructures show an enhancement of the coercive field, compared to $Co_{0.9}Fe_{0.1}$ grown on bare substrates, indicating a robust exchange coupling. The small exchange bias observed in our $Co_{0.9}Fe_{0.1}$/$BiFeO_3$ heterostructures is consistent with previous studies, which suggest that exchange bias is related to pinned uncompensated spins at $BiFeO_3$ domain walls[22,35,36]. The angular evolution of the remanent magnetization of the $Co_{0.9}Fe_{0.1}$ layers deposited on the $BiFeO_3$ obtained from the hysteresis loops are plotted in a polar curve (Fig. 5c). Similar to that observed in permalloy/$BiFeO_3$ single crystal structures wherein the spin-cycloid structure is present[37], the $Co_{0.9}Fe_{0.1}$ layers deposited on the $BiFeO_3$ thin films all present a uniaxial anisotropy; however, the coupling mechanism could be different since the spin cycloid has been suppressed in our films. Interestingly, for heterostructures grown on $SrTiO_3$, the exchange coupling between **L** and the ferromagnetic spin axis in the $Co_{0.9}Fe_{0.1}$ is strong enough to overcome the substrate-asymmetry and growth-field induced anisotropy and set the ferromagnetic spin along the $[1\bar{1}0]$. Since the in-plane projection of **L** on $SrTiO_3$ is along the $[00\bar{1}]$, one can conclude that the $BiFeO_3$ and $Co_{0.9}Fe_{0.1}$ spins are perpendicularly coupled. The perpendicular coupling at the antiferromagnetic/

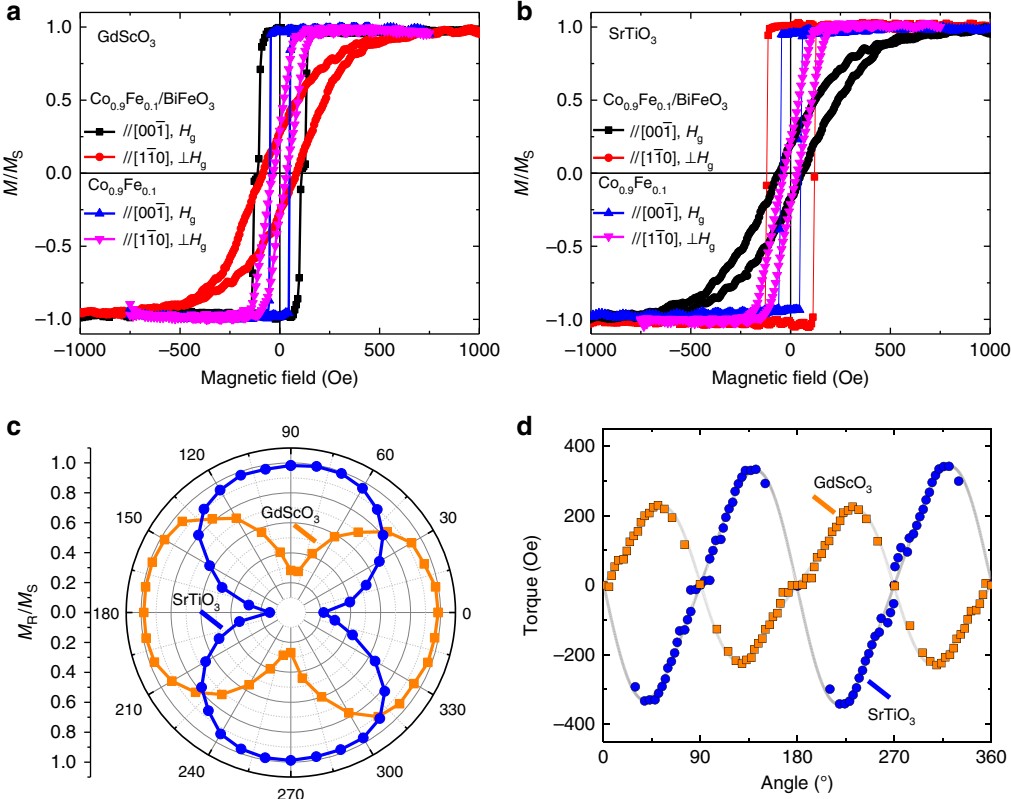

**Fig. 5** Controlling ferromagnetic anisotropy. **a** Room temperature magneto-optical Kerr effect (MOKE) hysteresis loops measured in-the-plane of the film for the Pt/Co$_{0.9}$Fe$_{0.1}$/BiFeO$_3$/GdScO$_3$ (010)$_O$ heterostructures with $H_g$ along [001] and similarly grown Pt/Co$_{0.9}$Fe$_{0.1}$/GdScO$_3$ (010)$_O$ heterostructures. **b** Room temperature MOKE hysteresis loops measured in-the-plane of the film for the Pt/Co$_{0.9}$Fe$_{0.1}$/BiFeO$_3$/SrTiO$_3$ (110) heterostructures with $H_g$ along [001] and similarly grown Pt/Co$_{0.9}$Fe$_{0.1}$/SrTiO$_3$ (110) heterostructures. **c** Polar plot of the evolution of $M_R/M_S$ for the Pt/Co$_{0.9}$Fe$_{0.1}$/BiFeO$_3$/GdScO$_3$ (010)$_O$ and SrTiO$_3$ (110) heterostructures. The angle is defined between the applied field $H$ and the [001]. **d** Magnetic torque as a function of angle of $M_R/M_S$ for the two heterostructures

$\Delta = 2.0$ eV; $U_{dd} = 5.0$ eV; $U_{dp} = 6.0$ eV; $10Dq = 0.8$ eV; $pd\sigma = -1.47$ eV; $pd\pi = 0.68$ eV; and the Slater integrals reduced to 70% of the Hartree–Fock values[29].

**Scanning transmission electron microscopy.** STEM work was performed in National Center for Electron Microscopy (NCEM), Molecular Foundry, Lawrence Berkeley National Laboratory. The samples for the STEM experiments were prepared by slicing, gluing, grinding, dimpling, and, finally, ion milling. Samples were subsequently Ar-ion milled using a Gatan Precision Ion Milling System II (PIPS II) with starting energies of 4 keV stepped down to an energy of 1 keV for the final milling. Before ion milling, the samples were dimpled down to less than 20 µm. High-angle annular dark field (HAADF)-STEM and bright-field (BF)-STEM imaging was performed on the Cs-corrected TEAM1 FEI Titan microscope at 300 kV. To enable determination of the atomic positions and Fe$^{3+}$ ion displacement vectors, noise in the HAADF images was filtered by Wiener filtering. The atom positions were determined accurately by fitting them as 2D Gaussian peaks by using a Matlab script[44]. The Fe$^{3+}$ ion displacement vector was calculated as a vector between each Fe$^{3+}$ and the center of mass of its two nearest neighbors Bi$^{3+}$.

**Co$_{0.9}$Fe$_{0.1}$ deposition and magnetic properties measurements.** For exchange-coupling studies, after the growth of the BiFeO$_3$ films, they were broken into two pieces and immediately inserted into a vacuum sputtering chamber with a base pressure of $8 \times 10^{-8}$ Torr. A 2.5 nm Pt/2.5 nm Co$_{0.9}$Fe$_{0.1}$ bilayer was deposited on the BiFeO$_3$ films and bare substrates by DC sputtering in $8 \times 10^{-4}$ Torr of Ar at room temperature with a 20 mT growth field along the in-plane [001] and [1$\bar{1}$0]. Magnetic hysteresis loops and magnetic anisotropy measurements were carried out using the longitudinal MOKE and rotation MOKE[45,46].

**Ab initio calculations.** Ab initio calculations were performed using the projector augmented wave (PAW)[47,48] formalism and a plane wave basis set, as implemented in the Vienna ab initio simulation package (VASP)[49,50]. The exchange and correlation potential were treated in the framework of generalized gradient approximation (GGA) of Perdew–Burke–Ernzerhof (PBE)[51]. The PAW potentials used explicitly treat 15 valence electrons for bismuth (5d$^{10}$ 6s$^2$ 6p$^3$), 14 for iron (3p$^6$ 3d$^6$ 4s$^2$), and 6 for

oxygen (2s$^2$ 2p$^4$). Local spin-density approximation with an additional Hubbard parameter (LSDA+U) was used for the exchange-correlation functional. The Hubbard parameter $U$ and the exchange interaction $J$ that treat the Fe $d$ electrons were set to $U = 2$ eV and $J = 0$ eV. Spin-orbit coupling (SOC) was included to calculate the non-collinear magnetic energy landscape. For the summation of charge densities over the Brillouin zone, a $3 \times 3 \times 3$ $k$-point mesh is adopted in the calculation of the total energy and force. The wave functions are expanded in plane waves up to a cutoff of 550 eV and the convergence precision of the total energy is set to be lower than $1 \times 10^{-6}$ eV. Symmetry was switched off to remove any artificial constraints on the possible spin ordering.

The supercell was made of $2 \times 2 \times 2$ cubic perovskite units, containing 40 atoms in total. Its three axes are [1$\bar{1}$0], [001], and [110], respectively, so as to simulate films of a (110)-oriented material (Supplementary Fig. S9). Under each strain (depicted by the misfit of crystal parameters), the in-plane [1$\bar{1}$0] and [001] axes of supercell are fixed while the out-of-plane [110] axis and the atomic positions are fully relaxed; a conjugate gradient algorithm is used and force precision is lower than 0.005 eV/Å. The polarization vector is evaluated by the Born effective charges using the Berry phase method.

The Hamiltonian containing exchange coupling, DMI, and SIA is written as:

$$H = -\sum_{ij} J_{ij}\mathbf{S}_i \cdot \mathbf{S}_j - \sum_{ij} \mathbf{D}_{ij} \cdot \mathbf{S}_i \times \mathbf{S}_j - K\sum_i |\mathbf{S}_i \cdot \mathbf{n}_i|^2 + H_0 \qquad (1)$$

where $\mathbf{S}_i$ is the $i$th spin vector; $J_{ij}$ is exchange parameter; $\mathbf{D}_{ij}$ is the DMI vector; $K$ is the SIA constant, and $\mathbf{n}_i = (\sin \theta_i \cos \varphi_i, \sin \theta_i \cos \varphi_i, \cos \theta_i)$ is the SIA unit vector in spherical coordinates. $H_0$ includes all other interactions, such as the lattice elastic energy. Here, we consider only the spin interaction between nearest neighbors. Three different DMI vectors $\mathbf{D}_1$, $\mathbf{D}_2$, and $\mathbf{D}_3$ are used for neighborhoods along $x$, $y$, and $z$ direction, respectively (Supplementary Fig. 12).

We used the method proposed by Xiang et al.[52] to calculate the DMI parameters. Specifically, to calculate $\mathbf{D}_1^x$, we considered four spin configurations, in which the spin of Fe$_1$ and Fe$_2$ are oriented along the $y$ and $z$ directions, respectively: (1) $S_1 = (0, S, 0)$, $S_2 = (0, 0, S)$; (2) $S_1 = (0, S, 0)$, $S_2 = (0, 0, -S)$; (3) $S_1 = (0, -S, 0)$, $S_2 = (0, 0, S)$; (4) $S_1 = (0, -S, 0)$, $S_2 = (0, 0, -S)$. The spin of the other six atoms are

the same and are along the $x$ direction: $S_{others} = (S, 0, 0)$. By computing the energy of the four spin configurations, we have:

$$\mathbf{D}_1^x = \frac{E_1 + E_4 - E_2 - E_3}{4S^2} \qquad (2)$$

The other two components of $\mathbf{D}_1$ and the vectors $\mathbf{D}_2$ and $\mathbf{D}_3$ can be computed similarly. The evolution of the three vectors under strain is plotted (Supplementary Fig. 14). Their average vector $\mathbf{D} = |\mathbf{D}_i| = (\mathbf{D}_1 + \mathbf{D}_2 + \mathbf{D}_3)/3$ is also plotted (Fig. 3b).

The SIA term is calculated by considering only the Fe ion with spin while all the other ions without spin. We performed constrained calculations with various directions of this isolated spin, so as to resolve the energy surface. The SIA constant $K$ is then fitted to these data points.

## Data availability

The data that support the findings of this study are available from the corresponding author upon reasonable request.

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

## Acknowledgements

The authors thank Prof. P. Yu for fruitful discussions. This work was funded by the U.S. Department of Energy, Office of Science, Office of Basic Energy Sciences, Materials Sciences and Engineering Division under Contract No. DE-AC02-05-CH11231 within the Materials Project program KC23MP for the synthesis of complex-oxide functional materials, the Non-Equilibrium Magnetic Materials program (MSMAG) for the first-principles calculations, and the van der Waals heterostructures program (KCWF16) for the magnetic studies of materials. This research used resources of the Advanced

Light Source, which is a DOE Office of Science User Facility under contract no. DE-AC02-05CH11231. STEM work at the Molecular Foundry was supported by the Office of Science, Office of Basic Energy Sciences, of the U.S. Department of Energy under Contract No. DE-AC02-05CH11231. This research used the resources of the National Energy Research Scientific Computing Center (NERSC) and Oak Ridge Leadership Computing Facility (OLCF) that are supported by the Office of Science of the U.S. Department of Energy, with the computational time allocated by the Innovative and Novel Computational Impact on Theory and Experiment (INCITEE) project. The study of multiferroic thin film devices was supported by the Army Research Office under grant W911NF-14-1-0104. L.R.D. acknowledges support U.S. Department of Energy, Office of Science, Office of Basic Energy Sciences under Award Number DE-SC-0012375. L.W.M. & R.R. acknowledge support from the Gordon and Betty Moore Foundation's EPiQS Initiative, Grant GBMF5307 and Intel, Corp. Z.H.C. acknowledges a startup grant from Harbin Institute of Technology, Shenzhen, China, under project number DD45001017.

## Author contributions

Z.H.C. conceived of and designed the research, and analyzed the results with R.R. and L. W.M. Z.H.C. and L.R.D. synthesized the samples. Z.H.C. performed the X-ray structural characterizations and PFM studies. The MOKE-based magnetic properties measurements were carried out by Z.H.C. with the assistance of Q.L., M.M.Y. and Z.Q.Q. Z.H.C., C.Y.K., A.F., L.Z. and C.K. performed the soft X-ray spectroscopy measurements with the assistance of P.S., E.A. and A.S., and interpreted the data with Z.W.H. and L.-H.T. Y.L.T. performed the STEM experiments and analyzed the data. Z.C. and L.-W.W. contributed to the theoretical studies. Z.H.C. and L.W.M. prepared the manuscript with the assistance of Z.C. and L.-W.W. All authors read and contributed to the manuscript and the interpretation of the data.

## Additional information

**Competing interests:** The authors declare no competing interests.

