## [Peer Review File · Nature Communications]

Reviewers' comments:

Reviewer #1 (Remarks to the Author):

The reported work deals with an interesting study of the antiferromagnetic spin structure of BiFeO₃ when submitted to different epitaxial strains. The studied systems are (110) oriented films of different thicknesses on two different substrates, so that compressive or tensile stress can be applied. Soft X-ray absorption spectroscopy measurements are used to analyze the direction of the antiferromagnetic vector L . Moreover, DFT calculations are performed in order to understand the origin of the measured evolution of the direction of L . It is shown that these are consistent with the calculated evolution of the Dzyaloshinskii-Moriya interaction vector as a function of the misfit strain. Finally, it is shown that exchange coupling with a ferromagnetic layer of CoFe presents a marked anisotropy perpendicular to L , thus hinting at an interface spin-flop coupling.

Although the manuscript is clear and the measurements interesting, I cannot recommend publication in Nat. Comm. as I think there is a major conceptual flaw in the understanding of the antiferromagnetic configuration of strain-free BiFeO₃. Indeed, beside exchange, the second important term in the free energy of pure unstrained BFO is the magneto-electric one, which is not even mentioned anywhere in the entire manuscript. This is epitomized by the paragraph on the properties of the bulk between lines 56 and 66. Unlike what is written, the L vectors in the incommensurate cycloidal state rotate in a plane containing P and are therefore not 'always perpendicular to P ' (line 65 + fig. 1a). This would be true indeed without the magneto-electric energy term which imposes (P,q) as the cycloidal plane (see for instance H. Katsura et al., PRL 95, 057205 (2005) or A. M. Kadomtseva et al., JETP Letters 79, 571 (2004)). In the same vein, the sentence lines 58 to 60 may be globally true for many oxides, but it is not for BFO. Indeed, in the bulk, the locally uncompensated moment due to the DMI term only warps the cycloid but remains averaged to zero because the magneto-electric term still imposes the incommensurate state. I also note that because the authors do not consider it in their calculations, they obtain a strain-free configuration where L is 'energetically degenerate within a plane perpendicular to P ', in complete disagreement with all experimental measurements. With the inclusion of the ME term, one could see the abrupt disappearance of the cycloidal state with progressive increase of strain. For a complete description of the competition between DMI and ME in BFO, I would advise the authors to read the abundant literature from the Russian team including A. Zvezdin (B. Ruetz, Phys. Rev. B 69, 064114 (2004) and especially its supplementary materials, A. M. Kadomtseva et al., JETP Letters 79, 571 (2004), as well as ref. 16 of this manuscript) or Randy S. Fishman, Phys. Rev. B 88, 104419 (2013) . The complete absence of even mentioning the magneto-electric coupling in the manuscript is by no means justified. Concerning the AF configuration at high anisotropy, the ME term loses its importance and BFO prefers the homogeneous state (above some value of anisotropy). Again, this is discussed in several papers from the Russian team, where it is also shown that the DMI interaction can be introduced in the free energy under the same form as the anisotropy. It is therefore already known that the orientation of L is determined by a competition between DMI and single ion anisotropy. Under this light, the conclusion of the paper is rather uneventful...

I also note that the authors always refer to the direction of L relative to that of P ('L is expected to be perpendicular to P'). I find this slightly misleading as I think the authors mean that 'L is expected to be perpendicular to [111]'. Comparing the directions of L and P seem to imply a direct coupling between these two quantities, which would thus be, by definition, magneto-electric in origin.

The experimental side of the manuscript is more impressive with the nice synchrotron vectorial measurements of L as well as the interesting exchange bias results. I have several questions regarding those.

1) It is written that the polarization vector P remains close to the [111] direction. This does not seem to be consistent with figure 3 where the model gives a 50% gradual decrease of the [110] projection of the polarization with strain.

2) The reported change in exchange anisotropy of the FeCo layer deposited on BFO films could be compared to similar measurements with single crystals where interface coupling was not invoked to be of the spin-flop type (see for instance M. Elzo et al., Phys. Rev. B 91, 014402 (2015)). Could the present results be consistent with this coupling model?

3) It is not clear if Exchange bias is indeed observed in the AF/F bilayers. If so, it seems rather small. How do the authors explain this behavior? As mentioned before, the DMI should induce some uncompensated magnetization, which in turn should couple to the ferromagnet and apply a bias. This is observed in BFO/F systems when BFO is in a homogeneous magnetic state, which is also the case here. The present results seem to indicate that exchange anisotropy dominates (like what happens for BFO bulk). A discussion concerning this point would be welcome.

To summarize, in spite of the beauty of some of the reported measurements, in my opinion the manuscript does not reach the standards for publication in Nature Communications, especially in view of the conceptual problems related to the unstrained state.

Reviewer #2 (Remarks to the Author):

Report on the manuscript, "Complex Strain Evolution of Polar and Magnetic Order and Control of Spin Orientation in Multiferroic BiFeO₃ Thin Films", by Zuhuang Chen et al. (manuscript number NCOMMS-17-34283-T).

This work reports on the fascinating result that the orientation of the antiferromagnetic axis in (110)-oriented BiFeO₃ films can be tailored from in-plane to out-of-plane by epitaxial strain. Interestingly, the authors find a deviation of the perpendicular relationship between the antiferromagnetic axis and the polarization vector. The results are topical, and the subject is well suited for publication in Nature Communications. However, I have some comments and concerns especially with respect to the analysis of the X-ray Linear Dichroism (XLD) data listed below. As these data are crucial for the interpretation mentioned above, the concerns should be dispelled in a revised version of the manuscript before I can recommend publication in Nature Communications. The comments and concerns are in detail:

1) The authors claim in the supplementary information in section III that the Fe L_{2,3} edge XLD is 'primarily of magnetic origin'. The authors support this statement by the Supplementary Figure 4a and 4b. Why do the authors only show the signal at the Fe L₂ edge in the Supplementary Figure 4b? As different final Fe 3d states are probed at the L₃ and L₂ edges, I am of the opinion that the presentation only at the L₂ edge is incomplete. Since the L₃ edge XLD is larger than the L₂ edge signal as seen in Fig. 1d of the main manuscript, the XLD (as the difference signal) at both edges (L₃ and L₂) must be presented throughout the manuscript. It is not sufficient to present the absorption coefficients for the different orientations of the E-vector. Coming back to the question of the origin of the XLD signal, I would be convinced if the authors present a much more detailed temperature-dependent study of the XLD especially in the vicinity of the Neel temperature. However, measurements only at two temperatures are shown, which actually exhibit a quite sizeable XLD signal at 600 K (Supplementary Figure 4b).

2) The authors write in the supplementary information in section III 'Normally, the magnetic linear dichroism gives rise to variations in peak intensity and the crystal-field effects give rise to shifts in peak position'. I think that this statement is incorrect as shown e.g. by early works on oriented molecules on surfaces (J. Stohr et al. Phys. Rev. Lett. 47, 381 (1981)). In this work the angular dependence is basically described by a variation of the peak intensity because of the different orbitals being probed at the respective orientation of the E-vector. Hence the statement above is not correct in general.

3) In the Supplementary Figure 5 again only results at the L₂ edge are presented. As already described at point 1), the results at both edges (L₃ and L₂) must be presented. The experimental results seem to exist as can be seen in Fig. 1d of the main manuscript and Supplementary Figure 4a. If the results of the atomic multiplet calculation have only been performed at the L₂ edge, then at least the experimental results must be shown at both edges.

4) Furthermore, in the Supplementary Figure 5 only the absorption coefficients for the different orientations of the E-vector with respect to the crystallographic orientation are shown. However, to allow for a critical inspection of the reader on the agreement between experiment and simulation the XLD as the difference signal (as e.g. shown Fig 1d) must be shown for all results of Supplementary Figure 5 a-l. Only this presentation would allow the reader to judge on the agreement of the XLD fine structures (at both L₃ and L₂ edges) with respect to intensity and energy position. By inspection by eye, I have the impression that the agreement with respect to the energy position of the first feature located at about 721.5 eV is not really excellent when comparing experiment and simulation. Therefore, the XLD must be shown as the difference signal for all spectra in the Supplementary Figure 5 a-l, maybe as an additional figure. Only this presentation would allow the reader to judge on the final interpretation of the orientation of L and the crystallographic axes in detail.

Reviewer #3 (Remarks to the Author):

It is a very interesting and important study. The subject is of increasing interest for the field of antiferromagnetic spintronics.

The authors investigated antiferromagnet BiFeO₃ which exhibits ferroelectricity and antiferromagnetism at room-temperature.

The state of the art experimental and theoretical methods were used in this work. The influence of epitaxial strain on the antiferromagnetic spin structure was revealed. They demonstrated that strain can drive a continuous reorientation of the antiferromagnetic spin axis from in-plane to out-of-plane directions.

The effect of the DMI and the single-ion anisotropy is revealed. I believe that results presented in this paper are novel and will promote new experimental

and theoretical studies of spin structure of antiferromagnets. The paper is well written.

I recommend publication of this paper in Nature Comm.

Overall Response to Editor and Reviewers

We thank the Editor and Reviewers for their time and effort in reviewing our manuscript. We would like to point out that all three Reviewers agreed that the present work/manuscript is clear, interesting, and systematic. Particularly, the enthusiasm for this work was made clear by Reviewers #2 and #3. In the following, we address and summarize our changes made in response to the suggestions/comments from the Reviewers and provide point-by-point responses to all queries. Additionally, changes to our manuscript are highlighted within the revised manuscript by using red text.

Reviewer #1:

Comment: “The reported work deals with an interesting study of the antiferromagnetic spin structure of BiFeO₃ when submitted to different epitaxial strains. The studied systems are (110) oriented films of different thicknesses on two different substrates, so that compressive or tensile stress can be applied. Soft X-ray absorption spectroscopy measurements are used to analyze the direction of the antiferromagnetic vector L . Moreover, DFT calculations are performed in order to understand the origin of the measured evolution of the direction of L . It is shown that these are consistent with the calculated evolution of the Dzyaloshinskii-Moriya interaction vector as a function of the misfit strain. Finally, it is shown that exchange coupling with a ferromagnetic layer of CoFe presents a marked anisotropy perpendicular to L , thus hinting at an interface spin-flop coupling.”

Response: We thank the Reviewer for his/her time in reviewing our manuscript and for the appreciation of the work as interesting. In the following, we attempt to carefully address any and all comments from the Reviewer.

Comments: “Although the manuscript is clear and the measurements interesting, I cannot recommend publication in Nat. Comm. as I think there is a major conceptual flaw in the understanding of the antiferromagnetic configuration of strain-free BiFeO₃. Indeed, beside exchange, the second important term in the free energy of pure unstrained BFO is the magneto-electric one, which is not even mentioned anywhere in the entire manuscript. This is epitomized by the paragraph on the properties of the bulk between lines 56 and 66. Unlike what is written, the L vectors in the incommensurate cycloidal state rotate in a plane containing P and are therefore not ‘always perpendicular to P ’ (line 65 + fig. 1a). This would be true indeed without the magneto-electric energy term which imposes (P,q) as the cycloidal plane (see for instance H. Katsura et al., PRL 95, 057205 (2005) or A. M. Kadomtseva et al., JETP Letters 79, 571 (2004)). In the same vein, the sentence lines 58 to 60 may be globally true for many oxides, but it is not for BFO. Indeed, in the bulk, the locally uncompensated moment due to the DMI term only warps the cycloid but remains averaged to zero because the magneto-electric term still imposes the incommensurate state. I also note that because the authors do not consider it in their calculations, they obtain a strain-free configuration where L is ‘energetically degenerate within a plane perpendicular to P ’, in complete disagreement with all experimental measurements. With the inclusion of the ME term, one could see the abrupt disappearance of the cycloidal state with progressive increase of stain.

For a complete description of the competition between DMI and ME in BFO, I would advise the authors to read the abundant literature from the Russian team including A. Zvezdin (B. Ruetter, Phys. Rev. B 69, 064114 (2004) and especially its supplementary materials, A. M. Kadomtseva et al., JETP Letters 79, 571 (2004), as well as ref. 16 of this manuscript) or Randy S. Fishman, Phys. Rev. B 88, 104419 (2013) . The complete absence of even mentioning the magneto-electric coupling in the manuscript is by no means justified.”

Response: We appreciate the recommendations and suggestions to improve the manuscript. We also apologize for the confusion that appears to have arisen concerning the nature of magnetism in BiFeO₃. In our original manuscript, the explanation of the magnetic spin structure for a strain-free single crystal was not sufficiently clear or detailed and thus did not sufficiently address the reality of the complex spin structure in BiFeO₃ single crystals. In reality, our current work is another, we believe important, example of the continuing evolution of our understanding of this complex material. For example, almost 30 years after the first observation of a cycloid spin structure in bulk BiFeO₃,¹ recent studies now report an even more complex spin structure in BiFeO₃ single crystals with an unexpected additional out-of-plane spin density wave that is superimposed on the (in-plane) magnetic cycloid with a small anharmonicity.^{2,3} As this illustrates, even in bulk BiFeO₃ the community’s understanding of the nature of this system continues to evolve. This is further complicated by the fact that there are additional clear difference between BiFeO₃ single crystals and thin-film versions,⁴⁻¹³ and thus the Reviewer may have potentially misunderstood some of the important aspects of the manuscript (as evidenced by some confusion over the nature of the discussions/results in our manuscript) and thus has the wrong impression of our understanding of the fundamental and basic nature of the spin structure in BiFeO₃. Due to space constraints in any paper, we could not provide a full recap of both the bulk and thin-film work on this subject to date. Alas, while there are of course many excellent works noted by the Reviewer, *our intention was to try to focus on the magnetic structure in BiFeO₃ thin films and not to provide a recap of the entire history and complex magnetic structure of BiFeO₃ overall.*² This was a decision based on a practical stance that to adequately first describe the bulk and then the evolution in thin films would have required considerable length and text that we didn’t believe we needed.

As such, we in no way meant to misrepresent the magnetic structure (which is rightly very complex) in bulk versions of BiFeO₃,² but to focus more on the thin-film systems where it is, again, different from the bulk. In this regard, we fully agree with the Reviewer’s point that the antiferromagnetic spin axis L in the incommensurate cycloidal state rotates in a plane containing P and therefore is not “always perpendicular to P .” To be clear, we did not state in our manuscript that the L is always perpendicular to the polarization vector P in bulk versions of BiFeO₃ with a spin-spiral structure, but were talking *only about thin-film versions*. We believe this is likely the single biggest misunderstanding about our work; of course, we take responsibility for that and for not addressing it clearly enough in our original manuscript. In the original manuscript, we do note (see lines 56-58) and as is shown in Fig. 8 and Fig. 13a in Ref. 10 by J.-G. Park *et al.*² (a comprehensive review paper about spin structure in bulk BiFeO₃ single crystals), that “The magnetic order of bulk BiFeO₃ is also complex, as it exhibits G-type antiferromagnetism with a superimposed long-wavelength cycloidal modulation along the $[1\bar{1}0]$.” This means that, the spins in bulk BiFeO₃ rotate within the $(11\bar{2})$, which is formed by the direction of the spontaneous polarization $[111]$ and the cycloid modulation vector $[1\bar{1}0]$.^{1,3,8,14,15} Our original point was that the $\{111\}$ are easy magnetization planes (and thus L is perpendicular to P , as shown in the schematic Fig. 1a) *in the absence of the cycloid spin order* (as is the case in our experimental results herein).

Such a structure has been shown in essentially all first-principles calculations (*e.g.*, from the works of N. Spaldin, L. Bellaiche, J. Íñiguez, etc.)^{4,11,16-19} and has been confirmed in epitaxial thin films in which the cycloid-spin structure is suppressed.^{5,6,9,11,13} Note that in the seminal work performed by Spaldin,⁴ no misfit strain was applied during the calculations, and it is found that the {111} are easy magnetization planes in BiFeO₃ thin films. Therefore, we mentioned in our original manuscript that “*L* is thought to always be perpendicular to *P*”, which refers only to BiFeO₃ thin films without the spiral-spin structure. We also note that, in the end, one of our main contributions in this manuscript is the demonstration that this perpendicular relationship can be broken by epitaxial strain.

Furthermore, we also agree that the incommensurate cycloid-spin structure would result in a cancellation of the weak ferromagnetic moment, but our intention here was to point out that the Dzyaloshinskii-Moriya interaction term would give rise to spin canting, thus resulting in a net weak ferromagnetic moment at the local scale. Such an observation has again been confirmed even in bulk-single crystals with cycloid-spin structure.³

What’s more, yet another misunderstanding may also exist in the discussion about the magnetoelectric coupling term and the Dzyaloshinskii-Moriya interaction. We fully agree with the Reviewer that the phenomenological approach, mostly performed by A. K. Zvezdin *et al.*,^{20,21} did attribute the origin of the cycloid order to the presence of the magnetoelectric-energy term (that is, the relativistic Lifshitz-like invariant). The Reviewer further suggested that the reason why our calculations did not reproduce the incommensurate cycloid state is simply because it did not include the magnetoelectric-energy term. In fact, our calculations *have considered magnetoelectric coupling already*, and the magnetoelectric coupling is essentially mediated by the spin-orbit based (*generalized*) Dzyaloshinskii-Moriya interaction at the atomic level.^{2,4,8,22-27} This is also pointed out by A. K. Zvezdin himself on page 2 of his work [*EPL* **99**, 57003 (2012)] where he states “*both the spin canting and the spin cycloid ordering stem from the same root, i.e., the Dzyaloshinskii-Moriya interaction*”.²⁸ This is further supported by other studies, for instance, in the work of J.-G. Park *et al.*, page 3 of *Phys. Rev. Lett.* **108**, 077202 (2012),²⁵ where it states that “*When we identify the antiferromagnetic vector in the Landau-Ginzburg theory of the local spin order parameter, we can show that the D term in our spin Hamiltonian is reduced to the same form of Lifshitz invariant as in JETP 79, 571 (2004) by A. M. Kadomtseva and A. K. Zvezdin et al.*”²¹ Further, in the second paragraph, page 17, in the review paper about spin structure in BiFeO₃ single crystal by J.-G. Park *et al.*, they further explicitly point out that “*The Lifshitz invariant was found to arise from the DM interaction*”;² and also on page 4 in the manuscript *Phys. Rev. Lett.* **109**, 067205 (2012) by M. Matsuda *et al.* where it states that “*Therefore, the cycloidal spin structure in BiFeO₃ is not caused by geometrical magnetic frustration as in many other multiferroic materials but rather by the relatively large DM interaction*”.²⁶ According to our understanding, the reason why all the previous first-principles calculations did not consider the cycloid spin structure in BiFeO₃ thin films is not simply because they did not include the magnetoelectric-energy term, but because of the extremely long period nature of the cycloidal spiral magnetic ordering (~62 nm) which would need huge computation resources to calculate such large cells and thus makes first-principles calculations of this cycloid-spin structure essentially untenable.

Furthermore, most experimental results on BiFeO₃ thin films, especially films with thicknesses less than ~70 nm, have found that the spiral-spin structure has been destroyed due to epitaxial constraint.^{5-7,9,11-13,29} For example, both work by the Berkeley team⁶ and the team lead by M. Bibes^{5,7} has reported that a simple *G*-type antiferromagnetic ordering without cycloid-spin structure in BiFeO₃ films as thick as even 300 nm grown on SrTiO₃ (001) and (111) substrates, in

which the misfit strain should be essentially fully relaxed. Further considering the observed large XMLD in all our films, all these facts further make the first-principles calculations using unrealistic huge cells to calculate the cycloidal spiral-spin structure unnecessary; therefore, most first-principles studies on BiFeO₃ assumed that the spiral-spin structure has been suppressed in BiFeO₃ thin films.^{4,11,16-19} Furthermore, as pointed by Zhang *et al.*,¹⁹ phenomenological approaches are believed to not be well suited for identifying the dominant microscopic mechanisms. In addition, our *ab-initio*-derived microscopic spin Hamiltonian is similar to previous studies,^{2,25,26,30-33} wherein such approaches are used to extract the magnetic anisotropy constant in single-crystal BiFeO₃ by J.-G. Park *et al.*,^{2,25,30} M. Matsuda *et al.*,²⁶ and R. S. Fishman *et al.*^{32,33} It is also prudent to highlight that these studies also did not include any specific magnetoelectric-coupling term in their spin Hamiltonian equation to derive/recapture the cycloid-spin structure. For instance, using inelastic neutron scattering, J.-G. Park *et al.* concluded that Dzyaloshinskii-Moriya interactions of nearest neighbor spins with $D \approx 0.1$ meV and its competition with superexchange (the symmetric exchange interaction favors a *G*-type antiferromagnetic ordering with antiparallel nearest-neighbor spins) is responsible for the cycloidal spiral-spin structure in bulk BiFeO₃.^{2,25,30} All told, microscopically, the spin structure in BiFeO₃ is thought to be determined by the interplay of exchange and Dzyaloshinskii-Moriya interactions and single-ion anisotropy, as suggested by several groups (*e.g.*, J.-G. Park, R. S. Fishman, N. Spaldin, S-W. Cheong, etc).^{2,25,26,30-36} Again, most of these studies are performed on BiFeO₃ single crystals or strain-free thin films; to the best of our knowledge, there are no studies on the microscopic origin of magnetic spin structure evolution in epitaxially constrained BiFeO₃ films. As such the current work, probing the strain evolution of the Dzyaloshinskii-Moriya interaction and the single-ion anisotropy and their effects on the magnetic anisotropy in BiFeO₃ films become important.

Overall, the Reviewer is right and his/her point is well taken – the spin structure and the physical mechanism underlying the formation of that spin structure in BiFeO₃ is complex. As suggested by the Reviewer and also to avoid further confusion, we have cited the references about BiFeO₃ listed by the Reviewer and have further modified the original sentences in and around lines 56-68 in the text to read:

“The magnetic order of bulk BiFeO₃ crystals is also complex, as it exhibits *G*-type antiferromagnetism with a superimposed long-wavelength cycloidal modulation along the $\langle 1\bar{1}0 \rangle$;^{1,2} that is, the spins rotate within the $\{11\bar{2}\}$ containing the direction of the spontaneous polarization P and the cycloid modulation vector.^{3,8} Antiphase oxygen octahedra rotations permit canting of the antiferromagnetic lattice through the Dzyaloshinskii-Moriya interaction (DMI) resulting in a local, weak canted moment M ,^{37,38} while the spin-cycloid structure results in cancellation of net macroscopic magnetization and linear magnetoelectric coupling in bulk BiFeO₃.^{21,31,39,40} This said, it is reported that epitaxial constraints in thin films can suppress the spin cycloid,^{5,7,13} and drive a transition toward a homogenous, weakly-ferromagnetic order with a preferred antiferromagnetic spin axis (L) in $\{111\}$, which is perpendicular to the oxygen octahedral rotation axis and the direction of P (Figure 1a).^{4,6,9,11} This can be viewed from a phenomenological Hamiltonian consisting of a DMI term and a spin-spin exchange interaction term.⁴ The DMI term has the form $E_{DM} = -D \cdot (L \times M)$, where E_{DM} is the DMI energy and D is the DM vector. Due to symmetry arguments, D is determined by the sense of rotation of the oxygen octahedra and is thus oriented along $\langle 111 \rangle$;⁴ *i.e.*, parallel to P . A perfect antiferromagnetic order as preferred by the exchange interaction term will have zero DMI energy, while a canting of

the magnetic moment can make the DMI energy negative, and the most efficient way to have such a canting and to reach the lowest energy is when L is perpendicular to D (and, in turn, P).⁴”

Comment: “Concerning the AF configuration at high anisotropy, the ME term loses its importance and BFO prefers the homogeneous state (above some value of anisotropy). Again, this is discussed in several papers from the Russian team, where it is also shown that the DMI interaction can be introduced in the free energy under the same form as the anisotropy. It is therefore already known that the orientation of L is determined by a competition between DMI and single-ion anisotropy. Under this light, the conclusion of the paper is rather uneventful...”

Response: We respectfully disagree with the Reviewer that the current findings of this work are uneventful. First, as we mentioned above, A. K. Zvezdin *et al.* (which, again, are now references in the updated text) theoretically explore the possible macroscopic mechanism of spin-structure evolution in BiFeO₃ through a phenomenological approach,^{10,20,21,40} while as pointed by Zhang *et al.*,¹⁹ the dominant microscopic mechanisms for the evolution of magnetic anisotropy in BiFeO₃ thin films is still unknown. Furthermore, indeed, there have been some previous studies on the role of Dzyaloshinskii-Moriya interactions and single-ion anisotropy in BiFeO₃ crystals, and we have and had cited these works in our original manuscript.^{2,26,30} To the best of our knowledge, however, *there is no systematic study on the microscopic mechanism of strain-driven spin structure change and how strain effects the Dzyaloshinskii-Moriya interaction and single-ion anisotropy in BiFeO₃ films, and, in turn, how these effects impact the antiferromagnetic-spin orientation.* Here, by using a combination of experimental studies and first-principles calculations, we complete what we believe is the first demonstration of the evolution of the Dzyaloshinskii-Moriya interaction and the single-ion anisotropy with epitaxial strain in BiFeO₃ thin films and explore the interplay of these terms in determining the antiferromagnetic-spin reorientation. As part of this, our work further reveals that the single-ion anisotropy is highly sensitive to the strain-induced lattice distortion and can come to compete with the Dzyaloshinskii-Moriya interaction with increasing strain; thus driving a deviation of the typically expected perpendicular relationship between L and P in films under tensile strain. This goes against previous studies where L and P are considered to remain perpendicular in BiFeO₃ thin films without the spiral-spin structure.^{4,11,16-19} In the end, we further demonstrate our understanding and control maps to a coupled ferromagnetic Co_{0.9}Fe_{0.1} layer which is strongly coupled to the BiFeO₃, and by engineering the antiferromagnetic-spin orientation, in turn, we can effectively tune the in-plane magnetic anisotropy of exchange-coupled Co_{0.9}Fe_{0.1} layers. Ultimately, we believe our studies enable a deeper understanding of the magnetic nature of BiFeO₃ thin films and exchange interaction at the BiFeO₃/ferromagnet interface.

Comment: “I also note that the authors always refer to the direction of L relative to that of P (L is expected to be perpendicular to P). I find this slightly misleading as I think the authors mean that ‘ L is expected to be perpendicular to [111]’. Comparing the directions of L and P seem to imply a direct coupling between these two quantities, which would thus be, by definition, magneto-electric in origin.”

Response: Indeed, as pointed out by the Reviewer, polarization P in bulk BiFeO_3 is along the $\{111\}$ and the reason why we refer to the direction of L with respect to P is that we want to emphasize the relationship between P and L and that the oriental relationship can be further modified by strain. The detailed reasons behind this are as follows: As we mentioned in the introduction, in most previous studies in BiFeO_3 without spiral-spin structure, either experimental works or theoretical studies, L is thought to be constrained within a $\{111\}$ due to the Dzyaloshinskii-Moriya interaction, and thus is perpendicular to P .^{4,6,11,13,16-19} As we demonstrate in the current work, however, we find that the perpendicular relationship between L and P can be broken by epitaxial strain due to the strain-induced enhancement of the single-ion anisotropy K which, while at zero strain is very small, can grow to compete in magnitude with the Dzyaloshinskii-Moriya interaction thereby driving a deviation of the perpendicular relationship between L and P . To further clarify, we note that magnetoelectric coupling could still exist in BiFeO_3 even though L and P are not perpendicular. In other words, L and P could be still correlated via the BiFeO_3 lattice itself such that L could be reoriented when P is rotated with an electric field because the whole BiFeO_3 unit cell may reorient. This is also reflected in previous work on bulk BiFeO_3 with a spiral-spin structure, even though P and L are not always perpendicular to each other in BiFeO_3 single crystals (as demonstrated by D. Lebeugle *et al.*⁸ and S. Lee *et al.*^{41,42}) and as is the BiFeO_3 thin films without spiral-spin structure,⁶ the antiferromagnetic ordering in the BiFeO_3 single crystal could still be manipulated by electric-field control of polarization and reorientation thereof.

Comment: “The experimental side of the manuscript is more impressive with the nice synchrotron vectorial measurements of L as well as the interesting exchange bias results. I have several questions regarding those. 1) It is written that the polarization vector P remains close to the $[111]$ direction. This does not seem to be consistent with figure 3 where the model gives a 50% gradual decrease of the $[110]$ projection of the polarization with strain.

Response: Indeed, as pointed out by the Reviewer, from the first-principle calculations, the out of plane polarization component $P_{[110]}$ decreases by $\sim 50\%$ when misfit strain varies from -10% to $+10\%$. Similar to that observed in (001) -oriented films in previous studies,⁴³ the in-plane misfit strain results in a rotation of the spontaneous polarization P within the $(1\bar{1}0)$ (Fig. R1). As suggested by the Reviewer, we carefully double checked the change in relative angle between P

Figure R1. The spontaneous polarizations P of the films under -10% , 0% , $+10\%$ misfit strains of BiFeO_3 . In-plane strains induce the rotation of the spontaneous polarization in the $(1\bar{1}0)$ plane.

and the [111]. In a strain-free film, P is along [111] and, using a Poisson ratio value obtained in BiFeO₃ films,⁴⁴ as the strain varies from a compressive strain of -10% to a tensile strain of +10%, the relative angle between the spontaneous polarization P and the [111] changes from +14° to -15° (Fig. R1); significantly less than the strain-induced change in L rotation. Following the Reviewer's suggestion, we have amended the text in the revised manuscript; it now reads: "This suggests that P rotates only by an amount of $\pm 15^\circ$ from the [111] within the (1 $\bar{1}$ 0)."

Comment: "(2) The reported change in exchange anisotropy of the FeCo layer deposited on BFO films could be compared to similar measurements with single crystals where interface coupling was not invoked to be of the spin-flop type (see for instance M. Elzo et al., Phys. Rev. B 91, 014402 (2015)). Could the present results be consistent with this coupling model?"

Response: This is an interesting suggestion. Indeed, the significant in-plane magnetic anisotropy observed in the ferromagnetic Co_{0.9}Fe_{0.1} layer grown on the (110)-oriented BiFeO₃ films is similar to that observed in permalloy (Py) layers grown on BiFeO₃ single crystals, as reported by M. Viret *et al.*,^{45,46} That is, the ferromagnetic magnetization is strongly coupled to the antiferromagnetic order in the BiFeO₃, either in single-crystal or thin-film form. As shown by M. Viret *et al.*,^{45,46} the spin cycloidal propagation vector in BiFeO₃ single crystals favors a parallel alignment with the ferromagnetic spins in Py. The coupling mechanism in Co_{0.9}Fe_{0.1}/BiFeO₃ film heterostructures like the ones studied herein, however, could be different from the Py/BiFeO₃ single-crystal structures since the spin cycloid has been suppressed in our films. From our XLD studies and exchange coupling studies, we found that, besides a clear enhancement of coercivity of the Co_{0.9}Fe_{0.1}, the ferromagnetic spins in the Co_{0.9}Fe_{0.1} are perpendicular to the antiferromagnetic spins in the BiFeO₃ – so not showing collinear coupling – which is consistent with previous reports of a spin-flop coupling mechanism being active.⁴⁷ Following the Reviewer's suggestion, we have amended the text in the revised manuscript and cited the work by M. Viret *et al.*, it now reads: "Similar to that observed in permalloy/BiFeO₃ single crystal structures wherein the spin-cycloid structure is present,^{45,46} the Co_{0.9}Fe_{0.1} layers deposited on the BiFeO₃ thin films all present a uniaxial anisotropy, however, the coupling mechanism could be different since the spin cycloid has been suppressed in our films."

Comment: "(3) It is not clear if Exchange bias is indeed observed in the AF/F bilayers. If so, it seems rather small. How do the authors explain this behavior? As mentioned before, the DMI should induce some uncompensated magnetization, which in turn should couple to the ferromagnet and apply a bias. This is observed in BFO/F systems when BFO is in a homogeneous magnetic state, which is also the case here. The present results seem to indicate that exchange anisotropy dominates (like what happens for BFO bulk). A discussion concerning this point would be welcome."

Response: Indeed, the exchange bias is rather small (< 15 Oe) in our heterostructures. As pointed by the Reviewer, the Dzyaloshinskii-Moriya interaction would give rise to a canted moment. The reason we did not observe a large exchange bias in our Co_{0.9}Fe_{0.1}/BiFeO₃ heterostructures is likely explained by a number of features: 1) The (110) surface of a G -type antiferromagnet is a compensated antiferromagnetic surface. 2) Our (110)-oriented BiFeO₃ films are single domain

and, from previous studies by the authors⁴⁸⁻⁵⁰ and also by M. Bibes *et al.*,⁷ exchange bias in BiFeO₃ has been found to be enhanced in the presence of pinned uncompensated spins at certain types of domain walls in BiFeO₃. As such, a rather small exchange bias is observed, although we do see an exchange enhancement of the coercive field of the ferromagnet; consistent with previous studies.^{7,48-50} In response to the Reviewer's comment, we have added a brief discussion and some citations to this point: "The small exchange bias observed in our Co_{0.9}Fe_{0.1}/BiFeO₃ heterostructures is consistent with previous studies which suggest that exchange bias is related to pinned uncompensated spins at BiFeO₃ domain walls."^{7,48-50}

Comment: "To summarize, in spite of the beauty of some of the reported measurements, in my opinion the manuscript does not reach the standards for publication in Nature Communications, especially in view of the conceptual problems related to the unstrained state."

Response: We hope we have illustrated above through both extensive descriptions and citations that some of the concerns and comments arose out of miscommunication. Having hopefully clarified those points, and strengthened the references and narration of the manuscript, we, in turn, believe we have produced a more clear and thorough presentation of our findings. Based on the collective quality of the work and the understanding and state of knowledge in the field, we believe that this work provides new physical insights critically important for understanding the magnetic structure of the most important multiferroic material BiFeO₃; topics which are both timely and exciting to the general readership of *Nature Communications*.

Reviewer #2:

Comment: "Report on the manuscript, "Complex Strain Evolution of Polar and Magnetic Order and Control of Spin Orientation in Multiferroic BiFeO₃ Thin Films", by Zuhuang Chen *et al.* (manuscript number NCOMMS-17-34283-T). This work reports on the fascinating result that the orientation of the antiferromagnetic axis in (110)-oriented BiFeO₃ films can be tailored from in-plane to out-of-plane by epitaxial strain. Interestingly, the authors find a deviation of the perpendicular relationship between the antiferromagnetic axis and the polarization vector. The results are topical, and the subject is well suited for publication in *Nature Communications*. However, I have some comments and concerns especially with respect to the analysis of the X-ray Linear Dichroism (XLD) data listed below. As these data are crucial for the interpretation mentioned above, the concerns should be dispelled in a revised version of the manuscript before I can recommend publication in Nature Communications. The comments and concerns are in detail:"

Response: We would like to thank the Reviewer for his/her strong support and for clearly pointing out the contributions of our while providing helpful comments to improve the manuscript. In the following, we will address each comment in detail and highlight changes/edits/addition to the text as required.

Comment: "1) The authors claim in the supplementary information in section III that the Fe *L*_{2,3} edge XLD is 'primarily of magnetic origin'. The authors support this statement by the

Supplementary Figure 4a and 4b. Why do the authors only show the signal at the Fe L_2 edge in the Supplementary Figure 4b? As different final Fe $3d$ states are probed at the L_3 and L_2 edges, I am of the opinion that the presentation only at the L_2 edge is incomplete. Since the L_3 edge XLD is larger than the L_2 edge signal as seen in Fig. 1d of the main manuscript, the XLD (as the difference signal) at both edges (L_3 and L_2) must be presented throughout the manuscript. It is not sufficient to present the absorption coefficients for the different orientations of the E-vector. Coming back to the question of the origin of the XLD signal, I would be convinced if the authors present a much more detailed temperature-dependent study of the XLD especially in the vicinity of the Neel temperature. However, measurements only at two temperatures are shown, which actually exhibit a quite sizeable XLD signal at 600 K (Supplementary Figure 4b).”

Response: We agree with the Reviewer that showing only the Fe- L_2 edge would be incomplete and it is necessary to include more temperature-dependent data points to further support the statement that the XLD in BiFeO₃ is dominated by magnetic origin. As we mentioned in the original manuscript, the magnetic contribution to the XLD can be determined by monitoring the linear dichroism as a function of temperature – the magnetic XLD is expected to decrease rapidly when the sample is heated to the Néel temperature (T_N). Following the Reviewer’s suggestion, we replotted the Fe XLD including both the L_2 and L_3 edges (Fig. R2). It can be seen that the difference between the two absorption coefficients for the different light polarizations, that is the XLD, reduces with increasing temperature, especially near the L_2 edge which essentially vanishes near T_N . In fact, the XLD at the Fe- L_2 edge at 600 K is $\sim 10\%$ of that at 300 K; therefore, we showed the XLD at the Fe- L_2 edge only in order to see the difference more clearly at the original manuscript. Following the Reviewer’s suggestion, additional temperature-dependent XLD measurements between 300 K and 650 K have been completed on a BiFeO₃/SrTiO₃ (110) heterostructure (Fig. R3). The XLD exhibits the characteristic shape at the L_2 edge (Fig. R3b) and decreases with temperature. At 650 K, *i.e.*, above T_N , the XLD at the L_2 edge essentially vanishes (Fig. R3c), indicating the purely magnetic origin of the observed dichroism; consistent with the recent studies by C.-H. Yang *et al.*⁵¹ and C. Y. Kuo *et al.*^{52,53} This is also further supported by our temperature-dependent XLD studies on (001)-oriented BiFeO₃ films grown on DyScO₃ (110)_o (Fig. R4) where the XLD, especially at the L_2 edge, is gradually reduced with increasing temperature, and becomes negligibly small near T_N .

Following the Reviewer’s suggestion, we now show XLD at both the Fe- $L_{2,3}$ edges as in Fig. R2 of the BiFeO₃ films grown on SrTiO₃ (110) and have added Fig. R3 in the Supplementary

Figure R2. a, XAS and b, XLD at the Fe $L_{2,3}$ edges at 300 K and 600 K for the BiFeO₃/SrTiO₃ (110) heterostructures.

Figure R3. Temperature dependent **a**, XAS and **b**, XLD at the Fe L_2 edges at different for a $\text{BiFeO}_3/\text{SrTiO}_3$ (110) heterostructure. The XLD ratio **c**, is normalized to the signal measured at 300K.

Materials. Furthermore, considering the fact that there still has some remnant XLD signal at the Fe- L_3 edge near T_N (even though it is much smaller than that at room temperature) and also in an attempt to be both conservative and true to our understanding, we have softened our language and make the change in the revised manuscript and supplementary materials, such that it reads: “In BiFeO_3 , temperature-dependent XLD studies have found that the intensity of the XLD signal near T_N is much smaller than at 300 K, especially for XLD at the Fe- L_2 edge which essentially vanishes (Supplementary Fig. 4 and Fig. 5); indicating that the XLD in BiFeO_3 is largely dominated by magnetic origin.” And further, “Temperature-dependent XLD at the Fe- $L_{2,3}$ edges for the BiFeO_3 films found that XLD near T_N is much smaller than that at 300 K (Supplementary Fig. 4b), especially for XLD at the L_2 edge which essentially vanishes at 650 K (Supplementary Fig. 5), indicating that the XLD in our BiFeO_3 heterostructures is largely dominated by magnetic origin.”

Figure R4. **a**, XAS and **b**, XLD at the Fe $L_{2,3}$ edges at 300 K and 620 K, and **c**, temperature dependent XAS at the Fe L_2 edge for the $\text{BiFeO}_3/\text{DyScO}_3$ (110)_o heterostructures.

Comment: “2) The authors write in the supplementary information in section III ‘Normally, the magnetic linear dichroism gives rise to variations in peak intensity and the crystal-field effects give rise to shifts in peak position’. I think that this statement is incorrect as shown e.g. by early works on oriented molecules on surfaces (J. Stohr *et al.* *Phys. Rev. Lett.* **47**, 381 (1981)). In this work the angular dependence is basically described by a variation of the peak intensity because of the different orbitals being probed at the respective orientation of the E-vector. Hence the statement above is not correct in general.”

Response: We thank the Reviewer for pointing this out. We agree that the statement about the angular dependence of XLD intensity is not valid in general, especially, in cases like Cu^{2+} , Mn^{3+} , and oriented molecules which have large orbital/crystal-field anisotropy.^{54,55} The strong orbital anisotropy will give rise to large crystal-field dichroism. In the case of high-spin Fe^{3+} of rhombohedral BiFeO_3 , however, which has a half-filled d^5 orbital with small orbital anisotropy, the polarization dependence with respect to the magnetization axis shows up as variations in the peak intensities, crystal-field dichroism gives rise to shifts in the peak position. As such, and following the Reviewer's suggestion, we have revised the manuscript as following: "In rhombohedral BiFeO_3 , which has a d^5 high-spin ground state for the Fe^{3+} , the magnetic linear dichroism mainly shows as variations in peak intensity,^{52,56} and the crystal-field effects typically give rise to shifts in peak position."⁵³

Comment: "3) In the Supplementary Figure 5 again only results at the L_2 edge are presented. As already described at point 1), the results at both edges (L_3 and L_2) must be presented. The experimental results seem to exist as can be seen in Fig. 1d of the main manuscript and Supplementary Figure 4a. If the results of the atomic multiplet calculation have only been performed at the L_2 edge, then at least the experimental results must be shown at both edges. 4) Furthermore, in the Supplementary Figure 5 only the absorption coefficients for the different orientations of the E-vector with respect to the crystallographic orientation are shown. However, to allow for a critical inspection of the reader on the agreement between experiment and simulation the XLD as the difference signal (as e.g. shown Fig 1d) must be shown for all results of Supplementary Figure 5 a-l. Only this presentation would allow the reader to judge on the agreement of the XLD fine structures (at both L_3 and L_2 edges) with respect to intensity and energy position. By inspection by eye, I have the impression that the agreement with respect to the energy position of the first feature located at about 721.5 eV is not really excellent when comparing experiment and simulation. Therefore, the XLD must be shown as the difference signal for all spectra in the Supplementary Figure 5 a-l, maybe as an additional figure. Only this presentation would allow the reader to judge on the final interpretation of the orientation of L and the crystallographic axes in detail."

Response: We thank the Reviewer for the suggestion. It is, indeed, necessary to include XAS and XLD at both the L_2 and L_3 edges. Following the Reviewer's suggestion, we provide more details by including XAS and XLD at both the L_2 and L_3 edges (Fig. R5) which is also now included as a figure in the Supplementary Materials of the revised manuscript. One can see that the calculations can reproduce in overall main features at both $L_{2,3}$ edge of the experimental spectral for BiFeO_3 film under various strain and AFM states.⁵⁷

Following the suggestion from the Reviewer and also considering the fact that the simulation is not 100% the same as the experimental result, we have softened our language in the revised Supplementary Materials, and replaced the previous text "One can see that the experimental spectra are nicely reproduced by the calculated spectra..." with "One can see that the calculations reproduce the main features at both the $L_{2,3}$ edges of the experimental spectral for the BiFeO_3 films under various strain and AFM states".

Figure R5. Experimental and calculated polarization-dependent Fe $L_{2,3}$ XAS and XLD spectra for **a-c**, 12 nm $\text{BiFeO}_3/\text{GdScO}_3$ (010)_o, **d-f**, 70 nm $\text{BiFeO}_3/\text{GdScO}_3$ (010)_o, **g-i**, 12 nm $\text{BiFeO}_3/\text{SrTiO}_3$ (110), and **j-l**, 70 nm $\text{BiFeO}_3/\text{SrTiO}_3$ (110) heterostructures with the incident beam parallel to the $[1\bar{1}0]$, $[001]$, and $[110]$.

Reviewer #3:

Comment: “It is a very interesting and important study. The subject is of increasing interest for the field of antiferromagnetic spintronics. The authors investigated antiferromagnet BiFeO_3 which exhibits ferroelectricity and antiferromagnetism at room-temperature. The state of the art experimental and theoretical methods were used in this work. The influence of epitaxial strain on the antiferromagnetic spin structure was revealed. They demonstrated that strain can drive a continuous reorientation of the antiferromagnetic spin axis from in-plane to out-of-plane directions. The effect of the DMI and the single-ion anisotropy is revealed. I believe that results presented in this paper are novel and will promote new experimental and theoretical studies of spin structure of antiferromagnets. The paper is well written. I recommend publication of this paper in Nature Comm.”

Response: We would like to thank the Reviewer for his/her strong support and important comments. We also highly appreciate that the Reviewer so clearly emphasizes that the “results presented in this paper are novel and will promote new experimental and theoretical studies of spin structure of antiferromagnets.”

References

- 1 Sosnowska, I., Peterlinneumaier, T. & Steichele, E. Spiral magnetic ordering in bismuth ferrite. *J. Phys. C: Solid State Phys.* **15**, 4835 (1982).
- 2 Je-Geun, P., Manh Duc, L., Jaehong, J. & Sanghyun, L. Structure and spin dynamics of multiferroic BiFeO₃. *J. Phys.: Condens. Matter.* **26**, 433202 (2014).
- 3 Ramazanoglu, M. *et al.* Local Weak Ferromagnetism in Single-Crystalline Ferroelectric BiFeO₃. *Phys. Rev. Lett.* **107**, 207206 (2011).
- 4 Ederer, C. & Spaldin, N. A. Weak ferromagnetism and magnetoelectric coupling in bismuth ferrite. *Phys. Rev. B* **71**, 060401 (2005).
- 5 Bea, H., Bibes, M., Petit, S., Kreisel, J. & Barthelemy, A. Structural distortion and magnetism of BiFeO₃ epitaxial thin films: A Raman spectroscopy and neutron diffraction study. *Philos. Mag. Lett.* **87**, 165-174 (2007).
- 6 Zhao, T. *et al.* Electrical control of antiferromagnetic domains in multiferroic BiFeO₃ films at room temperature. *Nat. Mater.* **5**, 823-829 (2006).
- 7 Béa, H. *et al.* Mechanisms of Exchange Bias with Multiferroic BiFeO₃ Epitaxial Thin Films. *Phys. Rev. Lett.* **100**, 017204 (2008).
- 8 Lebeugle, D. *et al.* Electric-field-induced spin flop in BiFeO₃ single crystals at room temperature. *Phys. Rev. Lett.* **100**, 227602 (2008).
- 9 Holcomb, M. B. *et al.* Probing the evolution of antiferromagnetism in multiferroics. *Phys. Rev. B* **81**, 134406 (2010).
- 10 Sando, D. *et al.* Crafting the magnonic and spintronic response of BiFeO₃ films by epitaxial strain. *Nat Mater* **12**, 641-646 (2013).
- 11 Heron, J. T. *et al.* Deterministic switching of ferromagnetism at room temperature using an electric field. *Nature* **516**, 370-373 (2014).
- 12 Ratcliff, W. *et al.* Neutron Diffraction Investigations of Magnetism in BiFeO₃ Epitaxial Films. *Adv. Funct. Mater.*, **21**, 1567–1574 (2011).
- 13 Bai, F. M. *et al.* Destruction of spin cycloid in (111)_c-oriented BiFeO₃ thin films by epitaxial constraint: Enhanced polarization and release of latent magnetization. *Appl. Phys. Lett.* **86**, 032511 (2005).

- 14 Sosnowska, I., Loewenhaupt, M., David, W. I. F. & Ibberson, R. M. Investigation of the unusual magnetic spiral arrangement in BiFeO₃, *Physica B* 180&181, 117–118 (1992).
- 15 Ramazanoglu, M. *et al.* Temperature-dependent properties of the magnetic order in single-crystal BiFeO₃. *Phys. Rev. B* **83**, 174434 (2011).
- 16 Albrecht, D. *et al.* Ferromagnetism in multiferroic BiFeO₃ films: A first-principles-based study. *Phys. Rev. B* **81**, 140401 (2010).
- 17 Wojdel, J. C. & Iniguez, J. Magnetoelectric Response of Multiferroic BiFeO₃ and Related Materials from First-Principles Calculations. *Phys. Rev. Lett.* **103**, 267205 (2009).
- 18 Picozzi, S. & Ederer, C. First principles studies of multiferroic materials. *J. Phys.-Condes. Matter* **21**, 303201 (2009).
- 19 Zhang, J. T. *et al.* Origin of magnetic anisotropy and spiral spin order in multiferroic BiFeO₃. *Appl. Phys. Lett.* **100**, 242413 (2012).
- 20 Sosnowska, I. & Zvezdin, A. K. Origin of the long period magnetic ordering in BiFeO₃. *J. Magn. Magn. Mater.* **140-144**, 167-168 (1995).
- 21 Kadomtseva, A., Zvezdin, A., Popov, Y., Pyatakov, A. & Vorob'ev, G. Space-time parity violation and magnetoelectric interactions in antiferromagnets. *JETP Letters* **79**, 571-581 (2004).
- 22 Katsura, H., Nagaosa, N. & Balatsky, A. V. Spin Current and Magnetoelectric Effect in Noncollinear Magnets. *Phys. Rev. Lett.* **95**, 057205 (2005).
- 23 Sergienko, I. A. & Dagotto, E. Role of the Dzyaloshinskii-Moriya interaction in multiferroic perovskites. *Phys. Rev. B* **73**, 094434 (2006).
- 24 Rahmedov, D., Wang, D., Íñiguez, J. & Bellaiche, L. Magnetic Cycloid of BiFeO₃ from Atomistic Simulations. *Phys. Rev. Lett.* **109**, 037207 (2012).
- 25 Jeong, J. *et al.* Spin Wave Measurements over the Full Brillouin Zone of Multiferroic BiFeO₃. *Phys. Rev. Lett.* **108**, 077202 (2012).
- 26 Matsuda, M. *et al.* Magnetic Dispersion and Anisotropy in Multiferroic BiFeO₃. *Phys. Rev. Lett.* **109**, 067205 (2012).
- 27 Tokunaga, M. *et al.* Magnetic control of transverse electric polarization in BiFeO₃. *Nat Commun* **6**, 6878 (2015).

- 28 Zvezdin, A. K. & Pyatakov, A. P. On the problem of coexistence of the weak ferromagnetism and the spin flexoelectricity in multiferroic bismuth ferrite. *EPL* **99**, 57003 (2012).
- 29 Bertinshaw, J. *et al.* Direct evidence for the spin cycloid in strained nanoscale bismuth ferrite thin films. *Nat. Commun.* **7**, 12664 (2016).
- 30 Jeong, J. *et al.* Temperature-Dependent Interplay of Dzyaloshinskii-Moriya Interaction and Single-Ion Anisotropy in Multiferroic BiFeO₃. *Phys. Rev. Lett.* **113**, 107202 (2014).
- 31 Fishman, R. S. Orientation dependence of the critical magnetic field for multiferroic BiFeO₃. *Phys. Rev. B* **88**, 104419 (2013).
- 32 Fishman, R. S., Haraldsen, J. T., Furukawa, N. & Miyahara, S. Spin state and spectroscopic modes of multiferroic BiFeO₃. *Phys. Rev. B* **87**, 134416 (2013).
- 33 Fishman, R. S., Furukawa, N., Haraldsen, J. T., Matsuda, M. & Miyahara, S. Identifying the spectroscopic modes of multiferroic BiFeO₃. *Phys. Rev. B* **86**, 220402 (2012).
- 34 Nagel, U. *et al.* Terahertz Spectroscopy of Spin Waves in Multiferroic BiFeO₃ in High Magnetic Fields. *Phys. Rev. Lett.* **110**, 257201 (2013).
- 35 Dixit, H., Hee Lee, J., Krogel, J. T., Okamoto, S. & Cooper, V. R. Stabilization of weak ferromagnetism by strong magnetic response to epitaxial strain in multiferroic BiFeO₃. *Scientific Reports* **5**, 12969 (2015).
- 36 Weingart, C., Spaldin, N. & Bousquet, E. Noncollinear magnetism and single-ion anisotropy in multiferroic perovskites. *Phys. Rev. B* **86**, 094413 (2012).
- 37 Dzialoshinskii, I. E. Thermodynamic Theory of "Weak" Ferromagnetism. In Antiferromagnetic Substances. *Soviet Physics JETP-USSR* **5**, 1259-1272 (1957).
- 38 Moriya, T. Anisotropic Superexchange Interaction and Weak Ferromagnetism. *Physical Review* **120**, 91-98 (1960).
- 39 Popov, Y. F. *et al.* Linear magnetoelectric effect and phase transitions in bismuth ferrite BiFeO₃. *JETP Letters* **57**, 69-73 (1993).
- 40 Ruelle, B. *et al.* Magnetic-field-induced phase transition in BiFeO₃ observed by high-field electron spin resonance: Cycloidal to homogeneous spin order. *Phys. Rev. B* **69**, 064114 (2004).

- 41 Lee, S. *et al.* Single ferroelectric and chiral magnetic domain of single-crystalline BiFeO₃ in an electric field. *Phys. Rev. B* **78**, 100101 (2008).
- 42 Lee, S., Ratcliff, W., Cheong, S. W. & Kiryukhin, V. Electric field control of the magnetic state in BiFeO₃ single crystals. *Appl. Phys. Lett.* **92**, 192906 (2008).
- 43 Jang, H. W. *et al.* Strain-Induced Polarization Rotation in Epitaxial (001) BiFeO₃ Thin Films. *Phys. Rev. Lett.* **101**, 107602 (2008).
- 44 Chen, Z. *et al.* Low-Symmetry Monoclinic Phases and Polarization Rotation Path Mediated by Epitaxial Strain in Multiferroic BiFeO₃ Thin Films. *Adv. Funct. Mater.* **21**, 133-138 (2011).
- 45 Lebeugle, D., Mougín, A., Viret, M., Colson, D. & Ranno, L. Electric Field Switching of the Magnetic Anisotropy of a Ferromagnetic Layer Exchange Coupled to the Multiferroic Compound BiFeO₃. *Phys. Rev. Lett.* **103**, 257601 (2009).
- 46 Elzo, M. *et al.* Coupling between an incommensurate antiferromagnetic structure and a soft ferromagnet in the archetype multiferroic BiFeO₃/cobalt system. *Phys. Rev. B* **91**, 014402 (2015).
- 47 Schulthess, T. C. & Butler, W. H. Consequences of Spin-Flop Coupling in Exchange Biased Films. *Phys. Rev. Lett.* **81**, 4516-4519 (1998).
- 48 Martin, L. W. *et al.* Nanoscale control of exchange bias with BiFeO₃ thin films. *Nano Letters* **8**, 2050-2055 (2008).
- 49 Chen, Z. *et al.* 180° Ferroelectric Stripe Nanodomains in BiFeO₃ Thin Films. *Nano Letters*, **15**, 6506 (2015).
- 50 Chen, D. *et al.* Interface Engineering of Domain Structures in BiFeO₃ Thin Films. *Nano Letters*, **17**, 5823 (2016).
- 51 Ko, K.-T. *et al.* Concurrent transition of ferroelectric and magnetic ordering near room temperature. *Nat Commun* **2**, 567 (2011).
- 52 Kuo, C. Y. *et al.* Single-domain multiferroic BiFeO₃ films. *Nat. Commun.* **7**, 12712 (2016).
- 53 Yang, J.-C. *et al.* Electrically enhanced magnetization in highly strained BiFeO₃ films. *NPG Asia Mater* **8**, e269 (2016).
- 54 Chen, C. T. *et al.* Out-of-plane orbital characters of intrinsic and doped holes in La_{2-x}Sr_xCuO₄. *Physical Review Letters* **68**, 2543-2546 (1992).

- 55 Stöhr, J., Baberschke, K., Jaeger, R., Treichler, R. & Brennan, S. Orientation of Chemisorbed Molecules from Surface-Absorption Fine-Structure Measurements: CO and NO on Ni(100). *Physical Review Letters* **47**, 381-384 (1981).
- 56 Kuiper, P., Searle, B. G., Rudolf, P., Tjeng, L. H. & Chen, C. T. X-ray magnetic dichroism of antiferromagnet Fe₂O₃: The orientation of magnetic moments observed by Fe 2*p* x-ray absorption spectroscopy. *Physical Review Letters* **70**, 1549-1552 (1993).
- 57 In calculations the crystal field parameters of *D*_{2h} symmetry for FeO₆ cluster are: 0 meV, -50 meV of Δe_g , and 0 meV, -10 meV of Δt_{2g} for BiFeO₃ films on GdScO₃ and SrTiO₃, respectively; 20 meV, -90 meV, 50 meV of *D*_u for the BiFeO₃ films on GdScO₃, the BiFeO₃ thinner film and thicker film on SrTiO₃, respectively; 50 meV, 100 meV of $E_x^2 - y^2 - z^2$ mix for BiFeO₃ films on GdScO₃ and SrTiO₃, respectively.

Reviewers' comments:

Reviewer #1 (Remarks to the Author):

The authors replied at some length to my comments, often providing too many details and repeating some already known properties of BFO (for instance concerning the magnetic state of bulk BFO). Although the manuscript is improved, there are still some conspicuously unclear points as the authors' answers are not all convincing. In particular, I do not fully understand the authors' point regarding the absence of cycloids in thin films with low strain. They underline the fundamental differences between the films and the bulk, which, to me, is rather surprising as in the work from Sando et al. (ref 10) cycloids are found stable in a quite large strain region. Moreover, I am a little worried by some contradictory comments in the authors' answer. For instance, in order to illustrate that films are very different from bulk BFO, it is argued that even in fully relaxed 300nm thick BFO films on STO(001) and (111) no cycloid can be found. At the same time, a central argument of the authors is that 'the spin spiral structure is destroyed by epitaxial constraint'. So, it is very unclear as to what makes the cycloid unstable in their films. This is rather central to the manuscript and as such, it deserves a discussion, especially since it does not seem consistent with the measurements of Sando et al.. Moreover, the authors state that the unstrained sample 'is thought to have L always perpendicular to P'. Further the authors write that 'one of our main contributions in this manuscript is the demonstration that this perpendicular relationship can be broken by epitaxial strain'. If this is so, I feel justified to maintain that this is a rather uneventful conclusion... It is indeed completely expected from the Zvezdin work on BFO (and common knowledge for magnetostriction). The direction of L in antiferromagnets is given by the direction of anisotropy which, for the bulk BFO case is along [111] for symmetry reasons. When adding strain along another direction, a different anisotropy constant enters the game along the direction of strain thus adding vectorially to the initial one (equivalently, the symmetry of the unit cell is modified). This obviously changes the resulting anisotropy direction away from [111]. Numerous studies on antiferromagnets (and ferromagnets of course) have already demonstrated this. What is of much higher calibre is the experimental demonstration of it in BFO, which is to me the strong point of the manuscript.

Concerning the modelling part of the paper, I understand then that the magnetoelectric effect is indeed accounted for in the calculations via the generalized DM interaction. I also understand that the cycloidal period is too long to be computed and therefore it is de facto assumed that the AF state is homogeneous (no cycloid). Therefore, the conclusion that L is perpendicular to P is a result of the assumption of a homogeneous AF state... This by no means explains the absence of cycloid in the authors' films.

Concerning the exchange coupling part, I think one can conclude that the magnetization of the FM couples parallel to the uncompensated moment in BFO. If this is correct, I think it should be stated as simply as this in the manuscript...

To conclude, although I find the manuscript improved, the answers to the few points I raised are only partly satisfying. The manuscript would be further improved by clarifying the points above.

Reviewer #2 (Remarks to the Author):

Second report on the revised manuscript, "Complex Strain Evolution of Polar and Magnetic Order and Control of Spin Orientation in Multiferroic BiFeO₃ Thin Films", by Zuhuang Chen et al. (manuscript number NCOMMS-17-34283-T).

In their revised version the authors dispelled all my concerns. I would just recommend multiplying the XLD differences in the Supplementary Figure 6 by a factor of two, i.e. to plot 'XLD difference x 2'. Thereby the similarities and the differences in the spectral features of the experimental and the theoretical XLD differences will be identified more clearly. The reason for this suggestion is that it is hard to analyze these signatures by eye in the current version. With these small modifications, I can recommend publication of the manuscript in Nature Communications.

Reviewer #1

Comment: “The authors replied at some length to my comments, often providing too many details and repeating some already known properties of BFO (for instance concerning the magnetic state of bulk BFO). Although the manuscript is improved, there are still some conspicuously unclear points as the authors’ answers are not all convincing. In particular, I do not fully understand the authors’ point regarding the absence of cycloids in thin films with low strain. They underline the fundamental differences between the films and the bulk, which, to me, is rather surprising as in the work from Sando *et al.* (ref 10) cycloids are found stable in a quite large strain region. Moreover, I am a little worried by some contradictory comments in the authors’ answer. For instance, in order to illustrate that films are very different from bulk BFO, it is argued that even in fully relaxed 300nm thick BFO films on STO(001) and (111) no cycloid can be found. At the same time, a central argument of the authors is that ‘the spin spiral structure is destroyed by epitaxial constraint’. So, it is very unclear as to what makes the cycloid unstable in their films. This is rather central to the manuscript and as such, it deserves a discussion, especially since it does not seem consistent with the measurements of Sando *et al.* Moreover, the authors state that the unstrained sample ‘is thought to have L always perpendicular to P ’. Further the authors write that ‘one of our main contributions in this manuscript is the demonstration that this perpendicular relationship can be broken by epitaxial strain’. If this is so, I feel justified to maintain that this is a rather uneventful conclusion... It is indeed completely expected from the Zvezdin work on BFO (and common knowledge for magnetostriction). The direction of L in antiferromagnets is given by the direction of anisotropy which, for the bulk BFO case is along [111] for symmetry reasons. When adding strain along another direction, a different anisotropy constant enters the game along the direction of strain thus adding vectorially to the initial one (equivalently, the symmetry of the unit cell is modified). This obviously changes the resulting anisotropy direction away from [111]. Numerous studies on antiferromagnets (and ferromagnets of course) have already demonstrated this. What is of much higher caliber is the experimental demonstration of it in BFO, which is to me the strong point of the manuscript.

Concerning the modelling part of the paper, I understand then that the magnetoelectric effect is indeed accounted for in the calculations via the generalized DM interaction. I also understand that the cycloidal period is too long to be computed and therefore it is de facto assumed that the AF state is homogeneous (no cycloid). Therefore, the conclusion that L is perpendicular to P is a result of the assumption of a homogeneous AF state... This by no means explains the absence of cycloid in the authors’ films.”

Response: To be clear, based on our results, it appears that the cycloidal-spin structure has been quenched in our films, and we think that the most likely reason is the substrate-induced epitaxial constraints. As was mentioned in the prior response (and we believe the Reviewer is already aware of this), most experimental results on BiFeO₃ films with thicknesses $\lesssim 70$ nm (which is the case in the current work) using various characterization techniques, have found that the cycloidal-spin structure has been destroyed and most of them have attributed the suppression of the spin cycloid to substrate-induced epitaxial constraint.¹⁻⁸ Also in the prior response, we noted that in the work of Béa *et al.*, *Phys. Rev. Lett.* **100**, 017204 (2008),² the cycloidal modulation was found to be suppressed in both 70 nm thick (001)-oriented and 300 nm thick (111)-oriented BiFeO₃ films grown on SrTiO₃ substrates using neutron diffraction. The specific selection and highlighting of this paper as an example was motivated by the desire to demonstrate and emphasize that the spin

structure of BiFeO₃ is highly sensitive to strain. It should be further noted that the 2008 *Phys. Rev. Lett.* we are discussing here originates from the same group as the Sando *et al.*, *Nature Mater.* **12**, 641-646 (2013) paper called out by the Reviewer⁹. As such, it must be possible that there is still some residual strain in the 300 nm thick (111)-oriented BiFeO₃ films, even though the residual strain is likely quite small; but apparently is still enough to induce the spin structure to change in the BiFeO₃ films.² This is further supported by the work of Bai *et al.*, *Appl. Phys. Lett.* **86**, 032511 (2005), where it was reported that a residual strain of <0.5% is enough to destroy the spin cycloid in 200 nm thick (111)-oriented BiFeO₃ films.³ Likewise, in Sando *et al.*, *Nature Mater.* **12**, 641-646 (2013), it is found that even a very small change of strain from -0.1% for the 70 nm thick BiFeO₃ films on GdScO₃ to +0.2% for the 70 nm thick BiFeO₃ films on SmScO₃ results in a significant change of the spin structure, *i.e.*, from bulk-like ‘type-1’ cycloid to a completely new ‘type-2’ cycloid that has not been observed in bulk before.⁹ In addition, in bulk BiFeO₃ crystals as studied in Ramazanoglu *et al.*, *Phys. Rev. Lett.* **107**, 067203 (2011), it is reported that a small uniaxial pressure of just 7 MPa (which corresponds to a uniaxial strain of only 0.01%) drives a large antiferromagnetic domain reorientation.¹⁰ These previous studies, from a range of groups around the world, suggest that the spin orientation/structure of BiFeO₃ is highly sensitive to strain and structural distortion. This strong sensitivity is likely related to the half-filled *d*⁵ orbital of Fe³⁺ in this material which has a negligibly small magneto-crystalline anisotropy;¹¹ therefore, similar to other ferrites such as α -Fe₂O₃,¹² even a small perturbation (arising, for instance, from strain) could be enough to drive a significant spin-structure change. Furthermore, because of the geometry difference (*i.e.*, large anisotropy in (110)-oriented films as compared with (001)-oriented films), the critical strain for driving the suppression of the spin cycloid in (001)-oriented films as reported by Sando *et al.*, could be different from the (110)-oriented films in the current work; something which has been suggested in previous studies.^{7,8}

As for the work noted by the Reviewer by Zvezdin, following the first round of reviews, we have already included calls to this work and use it to support our experimental results (in that our experimental results are consistent with the physical model presented therein). We further agree with the Reviewer that the change of the antiferromagnetic spin axis with strain is related to the magnetostrictive effect (which is indeed reported in other antiferromagnetic and ferromagnetic materials, as pointed out by the Reviewer). In fact, we would like to point out that the SIA and DMI are effectively the underlying microscopic mechanisms for the magnetostrictive effect in BiFeO₃. Our work, ultimately hones in these specific effects and the role they play in mediating the change in spin structure, not simply noting that it is related to the more generic magnetostrictive effect. In the end, our work is the first to experimentally realize and report such insights.

All told, we found that the antiferromagnetic spin structure in BiFeO₃ films is highly sensitive to epitaxial strain, and we think the absence of the spin cycloid in our BiFeO₃ films likely arises from the effect of substrate-induced epitaxial constraints which drive changes in the magnetic structure. As the antiferromagnetic spin axis is more sensitive to the strain than the ferroelectric polarization, the perpendicular relationship between *P* and *L* (normally imposed because of DMI) can be broken by epitaxial strain. Such an observation has not been reported in previous studies. More importantly, our work has provided the microscopic origin of the magnetic spin-structure evolution in epitaxially constrained BiFeO₃ films. We believe that this work provides new physical insights critically important for understanding the magnetic structure of the most important multiferroic material BiFeO₃; topics which are both timely and exciting to the general readership of *Nature Communications*.

Finally, following the Reviewer's suggestion, we have added discussion in our revised manuscript on page 11, where it now reads as: "The fine sensitivity of SIA to structural distortion may also explain how even small misfit strains in BiFeO₃ films can be sufficient to suppress the spin cycloid."

Comment: "Concerning the exchange coupling part, I think one can conclude that the magnetization of the FM couples parallel to the uncompensated moment in BFO. If this is correct, I think it should be stated as simply as this in the manuscript..."

Response: Indeed, based on our experimental results, the magnetization of the ferromagnet couples parallel to the uncompensated moment of BiFeO₃. Following this suggestion, we have added discussion on page 12 in our revised manuscript, where it now states: "The perpendicular coupling at the antiferromagnetic/ferromagnetic interface is due to the spin-flop coupling mechanism. According to the spin-flop mechanism, it is energetically preferred for a small canted moment in the antiferromagnet to couple parallel to the ferromagnetic magnetization, giving rise to a uniaxial magnetic anisotropy in the ferromagnet."

Comment: "To conclude, although I find the manuscript improved, the answers to the few points I raised are only partly satisfying. The manuscript would be further improved by clarifying the points above."

Response: We thank the Reviewer for raising multiple important and constructive points during the review processes. This feedback has been helpful in improving the manuscript. Despite what we consider to be a careful and detailed study here, we fully agree with the Reviewer that there could be still some remaining open questions as it pertains to the true physics of such systems which surely warrant further study. In the end, we hope that the Reviewer is pleased with the revised version of the paper.

Reviewer #2:

Comment: "Second report on the revised manuscript, "Complex Strain Evolution of Polar and Magnetic Order and Control of Spin Orientation in Multiferroic BiFeO₃ Thin Films", by Zuhuang Chen et al. (manuscript number NCOMMS-17-34283-T).

In their revised version the authors dispelled all my concerns. I would just recommend multiplying the XLD differences in the Supplementary Figure 6 by a factor of two, i.e. to plot 'XLD difference x 2'. Thereby the similarities and the differences in the spectral features of the experimental and the theoretical XLD differences will be identified more clearly. The reason for this suggestion is that it is hard to analyze these signatures by eye in the current version. With these small modifications, I can recommend publication of the manuscript in Nature Communications."

Response: We would like to thank the Reviewer for giving us very useful comments to improve the manuscript and for recommending it for publication. Following the Reviewer's suggestion, we have replotted Supplementary Figure 6 including the factor of two multiplication of the specific data. Please refer to Figure R1 below and also new Supplementary Fig. 6 in our revised manuscript.

Figure R1. Experimental and calculated polarization-dependent Fe $L_{2,3}$ XAS spectra and XLD for **a-c**, 12 nm BiFeO₃/GdScO₃ (010)_O, **d-f**, 70 nm BiFeO₃/GdScO₃ (010)_O, **g-i**, 12 nm BiFeO₃/SrTiO₃ (110), and **j-l**, 70 nm BiFeO₃/SrTiO₃ (110) heterostructures with the incident beam parallel to $[1\bar{1}0]$, $[001]$, and $[110]$.

References

- 1 Bea, H., Bibes, M., Petit, S., Kreisel, J. & Barthelemy, A. Structural distortion and magnetism of BiFeO₃ epitaxial thin films: A Raman spectroscopy and neutron diffraction study. *Philos. Mag. Lett.* **87**, 165-174, (2007).
- 2 Béa, H. *et al.* Mechanisms of Exchange Bias with Multiferroic BiFeO₃ Epitaxial Thin Films. *Phys. Rev. Lett.* **100**, 017204 (2008).
- 3 Bai, F. M. *et al.* Destruction of spin cycloid in (111)(c)-oriented BiFeO₃ thin films by epitaxial constraint: Enhanced polarization and release of latent magnetization. *Appl. Phys. Lett.* **86**, 032511 (2005).
- 4 Zhao, T. *et al.* Electrical control of antiferromagnetic domains in multiferroic BiFeO₃ films at room temperature. *Nature Mater.* **5**, 823-829 (2006).

- 5 Holcomb, M. B. *et al.* Probing the evolution of antiferromagnetism in multiferroics. *Phys. Rev. B* **81**, 134406 (2010).
- 6 Heron, J. T. *et al.* Deterministic switching of ferromagnetism at room temperature using an electric field. *Nature* **516**, 370-373 (2014).
- 7 Ratcliff, W. *et al.* Neutron Diffraction Investigations of Magnetism in BiFeO₃ Epitaxial Films. *Adv. Funct. Mater.*, 21, 1567–1574 (2011)
- 8 Bertinshaw, J. *et al.* Direct evidence for the spin cycloid in strained nanoscale bismuth ferrite thin films. *Nature Commun.* **7**, 12664 (2016).
- 9 Sando, D. *et al.* Crafting the magnonic and spintronic response of BiFeO₃ films by epitaxial strain. *Nature Mater.* **12**, 641-646 (2013).
- 10 Ramazanoglu, M. *et al.* Giant Effect of Uniaxial Pressure on Magnetic Domain Populations in Multiferroic Bismuth Ferrite. *Phys. Rev. Lett.* **107**, 067203 (2011).
- 11 Kuo, C. Y. *et al.* Single-domain multiferroic BiFeO₃ films. *Nature Commun.* **7**, 12712, (2016).
- 12 Artman, J. O., Murphy, J. C. & Foner, S. Magnetic Anisotropy in Antiferromagnetic Corundum-Type Sesquioxides. *Phys. Rev.* **138**, A912-A917, (1965).